# Generative Modeling of Discrete Latent Structures via Dynamic Policy Gradients

Stefan Ivanovic [1]   Ge Liu [1]   Mohammed El-Kebir [1 2]

## Abstract

Many scientific problems require inferring unobserved mechanistic latent states from indirect observations. While classical approaches, including expectation maximization, do not scale to combinatorially large spaces, deep learning approaches such as variational autoencoders typically form artificial latent states rather than reconstructing the mechanistic ground-truth states. Here, we introduce GReinSS, a policy learning framework that uses dynamically rescaled rewards to learn latent state distributions that maximize the observed data likelihood. We show that GReinSS accurately reconstructs simulated latent sets and latent graphs, outperforming alternative policy learning and generative modeling baselines. Additionally, GReinSS reconstructs isoforms from real short-read RNA sequencing data that better match isoforms detected by orthogonal long-read sequencing than the standard RSEM algorithm. Overall, GReinSS is a principled and practically effective approach for generative modeling and inference of combinatorial latent states from indirect observations.

## 1. Introduction

Learning latent states that explain observed data is a central goal of machine learning and probabilistic modeling. General-purpose unsupervised approaches such as clustering (MacQueen, 1965), topic modeling (Hofmann, 1999), and representation learning (Bengio et al., 2013) find *artificial latent states* that represent the variation in observed data $X_i$ without attempting to match some unknown true state $S_i^*$. On the other hand, practical and/or scientific problems often require inferring *mechanistic latent states* approxi-

mating unknown true states that are often combinatorial in nature, such as chemical reaction pathways (Ji & Deng, 2021), transportation networks (Biagioni & Eriksson, 2012), evolutionary trees (Felsenstein, 1981), molecular conformations (Noé et al., 2013), gene regulatory networks (Friedman et al., 2000), etc. Importantly, we do not observe the latent states $S_i$ and are instead given *indirect observations* $X_i$. These problems are self-supervised in the sense that the statistical process $\Pr(X \mid S)$ linking each indirect observation $X_i$ to each latent state $S_i$ is at least partially understood, rather than relying purely on unsupervised reconstruction of the observations $X_i$. Modeling mechanistic latent states can be formulated as optimizing a generative model $\Pr(S \mid \theta)$ with parameters $\theta$ that maximize the probability $\prod_i \sum_S \Pr(X_i \mid S) \Pr(S \mid \theta)$ of our observations $X_i$. The resulting inference problem is often solved using classical approaches such as expectation maximization (Dempster et al., 1977). However, classical approaches often struggle to scale to combinatorially large latent state spaces.

Reinforcement learning (RL) naturally fits combinatorial problems by sequentially generating their underlying structured latent states. While RL has been used to maximize expected rewards (Sutton et al., 1998), maximize entropy (Haarnoja et al., 2017), and match trajectory distributions to reward distributions (Malkin et al., 2022), and policy gradients have been used within variational inference to optimize surrogate objectives (Mnih & Rezende, 2016; Mnih & Gregor, 2014), these approaches do not directly optimize the marginal likelihood of indirect observations.

**Our contribution:** Here, we introduce Generative Reinforcement Learning of Structured States (GReinSS), a framework for optimizing the distribution $\Pr(S \mid \theta)$ of states $S$ to maximize the overall probability $\prod_i \Pr(X_i \mid \theta)$ of observations $X_i$. GReinSS uses policy gradients with dynamically scaled rewards to ensure that the policy's parameters $\theta$ are updated in the direction of maximizing the probability $\prod_i \Pr(X_i \mid \theta)$ of the observations $X_i$. In contrast to standard RL formulations, where policy gradients optimize expected return under a stationary reward function, GReinSS uses RL machinery as an optimization tool for probabilistic modeling over discrete latent states.

We compare GReinSS with existing approaches on both simulated latent graph inference, simulated latent set re-

[1]Siebel School of Computing and Data Science, University of Illinois at Urbana-Champaign, IL 61801, USA [2]Cancer Center at Illinois, University of Illinois Urbana-Champaign, IL 61801, USA. Correspondence to: Mohammed El-Kebir <melkebir@illinois.edu>.

*Proceedings of the 43rd International Conference on Machine Learning*, Seoul, South Korea. PMLR 306, 2026. Copyright 2026 by the author(s).

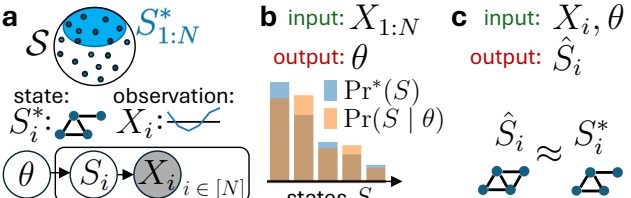

*Figure 1.* **Overview of GReinSS. a** Many scientific problems consist of estimating latent states $S_1^*, \ldots S_N^* \in \mathcal{S}$ only given indirect observations $X_{1:N} = X_1, \ldots, X_N$. These latent states are sampled from an unobserved underlying distribution $\Pr^*(S)$, which we model as $\Pr(S \mid \theta)$ using parameters $\theta$. This yields two problems. **b** First, a learning problem of identifying parameters $\theta$ that maximize the data likelihood $\Pr(X_{1:N} \mid \theta)$. **c** Second, an inference problem of estimating $S_i^*$ with $\hat{S}_i = \text{argmax} \Pr(S \mid X_i) \Pr(S \mid \theta)$ given $X_i$ and $\theta$. GReinSS solves both problems using policy gradients with dynamic rewards.

construction, and the real biological problem of RNA isoform reconstruction from short-read sequencing data. On simulations, GReinSS reliably reconstructs ground-truth latent states with higher accuracy than standard policy gradients, GFlowNets, local search, and generalized expectation maximization implemented with variational autoencoders, diffusion models, and autoregressive models. On the real short-read RNA sequencing data from GTEx (Lonsdale et al., 2013), GReinSS outperforms the standard RSEM (Li & Dewey, 2011) method used by GTEx in terms of predicting gene isoforms validated by additional long-read RNA sequencing. These results demonstrate that GReinSS successfully extends modern generative modeling to the inference of discrete latent states from indirect observation data through a policy learning formulation.

### 1.1. Related Works

There are several related works aimed at explaining observation data $X_1, \ldots, X_N$ through latent states $S$. The simplest approach for inferring a latent state $S$ from an observation $X_i$ given the probability function $\Pr(X_i \mid S)$ is directly optimizing $\text{argmax}_S \Pr(X_i \mid S)$ via local search. This, however, does not leverage information across observations $X_i$ through a shared model $\Pr(S \mid \theta)$ of latent state generation.

Variational autoencoders effectively learn a distribution over artificial latent states that maximizes the probability of generating observation data $X_1, \ldots, X_N$ (Kingma & Welling, 2013). However, these artificial latent states belong to a completely separate vector space distinct from the mechanistic latent states that underlie the scientific problem.

The classical approaches of expectation maximization (EM) and generalized expectation maximization (GEM) for inferring problem-specific mechanistic latent states/variables $S$ operate by alternating between inference over latent variables $S$ given parameters $\theta$ (E-step) followed by updat-

ing parameters $\theta$ given latent variables $S$ (M-step) (Dempster et al., 1977; Wu, 1983). This can be highly effective but requires the ability to compute the expectation value $\sum_{i=1}^{N} \mathbb{E}_{S_i \sim \Pr(S \mid X_i, \theta^{(t)})}[\log(\Pr(X_i, S_i \mid \theta^{(t+1)}))]$ of the complete-data log-likelihood. In settings where the latent state space is exponentially large, calculating this expectation is generally intractable. A classical exception is hidden Markov models, where the exponentially large state space has a Markov structure that allows exact expectation computation via dynamic programming (Rabiner, 1989). In Section 3.3, we describe how GEM can be adapted to use modern machine learning methods, including variational autoencoders, autoregressive models, and discrete diffusion, to approximately compute and optimize this expectation value on general exponentially large state spaces. More recently, variational inference (VI) (Wainwright et al., 2008) has emerged as a principled framework for learning probabilistic models on exponentially large latent spaces. Rather than directly optimizing the marginal likelihood $\Pr(X_{1:N} \mid \theta)$ using a known distribution $\Pr(X \mid S)$, VI solves a related problem in which the objective is replaced by an ELBO surrogate objective that includes an explicit variational posterior model $q_\phi(S \mid X)$ parameterized by $\phi$.

Beyond EM and VI, reinforcement learning has been widely used for generating structured data using discrete action spaces (Yu et al., 2017). For instance, GFlowNets are used to learn policies that generate states $S$ with probabilities $\Pr(S \mid \theta)$ proportional to their reward (Malkin et al., 2022). However, this method assumes known rewards for terminal states $S$ rather than optimizing the distribution of terminal states to maximize the probability $\Pr(X_1, \ldots, X_N \mid \theta)$ of indirect observation data. Many approaches have used adaptive, normalized, or shaped reward functions; however, these are typically used to encourage exploration or stabilize training rather than to match some latent distribution of states (Ibrahim et al., 2024; Yuan et al., 2023). GReinSS instead constructs adaptive rewards that enable policy gradients to learn a distribution $\Pr(S \mid \theta)$ over latent states $S$ that maximizes the likelihood $\Pr(X_1, \ldots, X_N \mid \theta)$ of indirect observation data $X_1, \ldots, X_N$. While the aforementioned methods do not generally solve the same problem as GReinSS, we identify several special cases where they do (Section 3.3). Moreover, we use these methods as baselines in our evaluation (Section 4.1).

In addition to related general methods, two previous papers have utilized the underlying technique of GReinSS without generalizing to describe the full GReinSS procedure (Ivanovic & El-Kebir, 2023; 2025) (see Section B.4 for details). We build upon these early applications and (i) formulate general probabilistic learning and inference problems on arbitrary discrete structures (Section 2); (ii) provide a theoretically grounded solution to these problems by establishing a connection between policy-gradient training and

maximum likelihood estimation in discrete combinatorial spaces (Section 3.1); (iii) formulate a theory for optimal off-policy sampling (Section 3.2); (iv) relate existing ML approaches to GReinSS (Section 3.3); and (v) provide ablations and comparisons with existing ML methods (Section 4).

**Conflict of Interest Disclosure:** None.

## 2. Problem Definition

Let $\Pr^*(S)$ be the ground-truth distribution on the set $\mathcal{S}$ of latent states $S$, which can represent graphs, sequences, sets, positions in space, or any discrete data structure that can be generated via policy learning. We are given a sample of $N$ items from this distribution. Importantly, we do not directly observe the latent states $S_{1:N}^* = S_1^*, \ldots, S_N^*$; instead we are only given indirect observations $X_{1:N} = X_1, \ldots, X_N$. In addition, we are given the ability to compute the probability $\Pr(X \mid S)$ of any observation $X$ given any latent state $S$. As such, our observations $X_{1:N}$ are generated from $\Pr(X \mid S_1^*), \ldots, \Pr(X \mid S_N^*)$. We aim to approximate $\Pr^*(S)$ via a generative model $\Pr(S \mid \theta)$ of latent states $S$ with parameters $\theta$ (Figure 1a). We note that we place no restrictions on the generative model, which can be a variational autoencoder, discrete diffusion, autoregressive generation, policy-based generation, etc. Therefore, the probability of an observation $X$ given parameters $\theta$ is

$$\Pr(X \mid \theta) = \sum_{S \in \mathcal{S}} \Pr(X \mid S) \Pr(S \mid \theta). \qquad (1)$$

The overall probability $\Pr(X_{1:N} \mid \theta)$ of the observations $X_{1:N}$ given our model parameters $\theta$ thus equals $\prod_{i=1}^{N} \sum_{S \in \mathcal{S}} \Pr(X_i \mid S) \Pr(S \mid \theta)$. We pose the following learning problem to train $\Pr(S \mid \theta)$ towards matching $\Pr^*(S)$ without directly observing $\Pr^*(S)$ (Figure 1b).

**Problem 2.1** (LATENT STATE MODELING FROM INDIRECT OBSERVATIONS). Given indirect observations $X_{1:N}$ and the probability function $\Pr(X \mid S)$, find model parameters $\theta$ that maximize the probability $\Pr(X_{1:N} \mid \theta)$ of the observations.

Next, we pose the following inference problem to estimate the unobserved ground-truth latent states $S_1^*, \ldots, S_N^*$ using the learned parameters $\theta$ (Figure 1c).

**Problem 2.2** (LATENT STATE INFERENCE). Given indirect observations $X_{1:N}$, the probability function $\Pr(X \mid S)$, and model parameters $\theta$, for each observation $X_i$ find latent state $\hat{S}_i \in \mathcal{S}$ that maximizes the probability $\Pr(\hat{S}_i \mid X_i, \theta) = \Pr(X_i \mid \hat{S}_i) \Pr(\hat{S}_i \mid \theta)$.

Problems 2.1 and 2.2 also apply in more complex situations with distinct groups of observations and states. For instance, if multiple pieces of data are generated from each state $S_i^*$, one can group these *sub-observations* into one full observation $X_i$ (see Section B.1 and Section 4.1). Additionally, multiple environments $v$, each with (i) their own distributions $\Pr_v^*(S)$ of latent states, (ii) lists $S_{v,1}^*, \ldots S_{v,N^v}^*$ of ground-truth latent states, (iii) distributions $\Pr_v(X \mid S)$ of observations, and (iv) lists $X_1^v, \ldots X_{N^v}^v$ of observations, can be subsumed in the original problem definitions (see Section B.1 and Section 4.2).

## 3. Methods

Proofs for theorems in this section are provided in Section A.

### 3.1. The GReinSS Method

We solve the learning problem (Problem 2.1) of identifying parameters $\theta$ maximizing $\Pr(X_{1:N} \mid \theta)$ via gradient descent by estimating the gradient $\frac{d}{d\theta} \log(\Pr(X_{1:N} \mid \theta))$ of the data log-likelihood. In the following, we show how to accomplish this using policy gradients and a dynamically changing reward function.

We define $\theta$ as the parameters of our policy. The policy defines a probability distribution $\Pr(\tau \mid \theta)$ over trajectories $\tau$, which are sequences of actions concluding with a terminal state $S(\tau)$. In our case, the terminal state $S(\tau)$ of each trajectory $\tau$ corresponds to a latent state in $\mathcal{S}$. As such, we define $\Pr(X \mid \tau) = \Pr(X \mid S(\tau))$. Analogous to Equation (1), we define

$$\Pr(X \mid \theta) = \mathbb{E}_\tau[\Pr(X \mid \tau)] = \mathbb{E}_\tau[\Pr(X \mid S(\tau))] \quad (2)$$

where $\tau$ is drawn from the distribution $\Pr(\tau \mid \theta)$. In practice, we estimate the quantity $\Pr(X_i \mid \theta)$ by sampling, i.e., $\Pr(X_i \mid \theta) \approx \frac{1}{M} \sum_{j=1}^{M} \Pr(X_i \mid \tau_j)$ where $\tau_1, \ldots \tau_M$ are sampled from $\Pr(\tau \mid \theta)$. This leads us to the following important theorem.

**Theorem 3.1.** *The policy gradient $\mathbb{E}_\tau[r(\tau) \frac{d}{d\theta} \log(\Pr(\tau \mid \theta))]$ with dynamically changing rewards*

$$r(\tau) = \sum_{i=1}^{N} \frac{\Pr(X_i \mid \tau)}{\Pr(X_i \mid \theta)} \qquad (3)$$

*is an unbiased estimator of the gradient $\frac{d}{d\theta} \log(\Pr(X_{1:N} \mid \theta))$ of the log-likelihood objective.*

Intuitively, the denominator $\Pr(X_i \mid \theta)$ rescales each observation's contribution to the total reward such that trajectories are rewarded based on their proportional contribution to $\Pr(X_i \mid \theta)$ rather than the raw probability $\Pr(X_i \mid \tau)$. This rescaling results in solving for the optimal distribution over trajectories $\Pr(\tau \mid \theta)$ rather than converging to one highest reward trajectory as illustrated in Section B.2 and Figure S1. As such, we can apply standard policy gradient updates to solve Problem 2.1.

**Corollary 3.2.** *GReinSS's application of standard policy gradient updates to $\theta$ using the reward $r(\tau)$ performs gradient ascent on the data log-likelihood* $\log(\Pr(X_{1:N} \mid \theta))$.

In other words, after each gradient update yielding parameters $\theta$, GReinSS updates the dynamic rewards $r(\tau)$ using these new parameters via (2) and (3), which in turn are used for the next gradient update of $\theta$. We note that in standard RL, policy gradients $\mathbb{E}_\tau[r(\tau)\frac{d}{d\theta}\log(\Pr(\tau \mid \theta))]$ are used to maximize the expectation value $\mathbb{E}_\tau[r(\tau)]$ of the rewards, given a fixed reward function $r(\tau)$ independent of the policy parameters $\theta$. Our dynamic rewards, which depend on parameters $\theta$, allow the same policy gradients procedure to instead optimize the data log-likelihood $\log(\Pr(X_{1:N} \mid \theta))$ of our observations $X_{1:N}$. Importantly, despite the rewards being calculated using $\theta$, the gradient $\frac{d}{d\theta}$ is only applied to $\log(\Pr(\tau \mid \theta))$ not $r(\tau)$.

After optimizing $\Pr(X_{1:N} \mid \theta)$ to solve Problem 2.1, one can solve Problem 2.2 by sampling latent states and approximating which latent state $\hat{S}_i$ maximizes $\Pr(\hat{S}_i \mid \theta)\Pr(X_i \mid \hat{S}_i)$ for each data point $i$ (details in Section B.5).

Although Theorem 3.1 calculates the reward over all observations, policy gradients still give an unbiased estimator of the log-likelihood objective if one applies mini-batching to calculate rewards (Theorem A.1). Specifically, for a mini-batch $B \subset [N]$ the mini-batch reward is $r_B(\tau) = \sum_{i \in B} \Pr(X_i \mid \tau)/\Pr(X_i \mid \theta)$. Analogous to mini-batch stochastic gradient ascent in standard likelihood-based learning, applying mini-batching to GReinSS can enable scaling to very large datasets.

## 3.2. Off-policy Learning and Importance Sampling

In many cases, off-policy learning is vital for practically solving Problem 2.1. Specifically, sampling from our policy directly may generate very few trajectories $\tau$ that give a high (or even nonzero) probability $\Pr(X_i \mid \tau)$ for some (or even all) observations $X_i$. Thus, it makes sense to sample from a distribution that explicitly takes into account $X_{1:N}$ and ensures the sampling of latent states $S$ that have reasonably high probabilities $\Pr(X_i \mid S)$ for various observations $X_i$. To this end, we use Bayes' theorem and obtain $\Pr(\tau \mid X_i, \theta) = \Pr(X_i \mid \tau)\Pr(\tau \mid \theta)/\Pr(X_i \mid \theta)$.

We then have the following theorem.

**Theorem 3.3.** *The unbiased variance-minimizing off-policy sampling proposal is* $q(\tau \mid X_{1:N}, \theta) = \frac{1}{N}\sum_{i=1}^N \Pr(\tau \mid X_i, \theta)$.

Generally, one cannot directly sample from $q(\tau \mid X_{1:N}, \theta)$ but can use heuristics to bias the sampling towards $q(\tau \mid X_{1:N}, \theta)$. For example, in the cancer phylogeny application CloMu (Ivanovic & El-Kebir, 2023), the observations $X_i$ directly show which nodes are in the graph generated by

*Table 1.* **Comparison of GReinSS to baseline methods.** While local search can infer individual latent states $S_i$ from each observation $X_i$, it does not utilize parameters $\theta$. While variational autoencoders (VAE), autoregression, and diffusion can be adapted to solve Problem 2.1 via Generalized Expectation-Maximization (GEM), it requires an inexact modification of the GEM algorithm. On the other hand, the two policy learning (PL) baselines maximize proxy objectives.

| METHOD | PARADIGM | MAXIMIZES OBJ. $\Pr(X_{1:N} \mid \theta)$ |
|---|---|---|
| LOCAL SEARCH | SEARCH | NO |
| VAE | GEM | APPROX. |
| AUTOREGRESSION | GEM | APPROX. |
| DIFFUSION | GEM | APPROX. |
| NAIVE POLICY GRAD. | PL | NO |
| GFLOWNETS | PL | NO |
| GREINSS | PL | YES |

$\tau$. Thus, the off-policy sampling procedure only allows trajectories that generate a set of nodes corresponding to some observation $X_i$ in $X_{1:N}$. As another example, CN-Rein (Ivanovic & El-Kebir, 2025) first uses a simple non-machine learning algorithm called CNNaive to give initial estimates of plausible latent states and then biases trajectories towards these plausible latent states. After applying off-policy learning, one can account for this in policy gradients using importance sampling (Section B.6).

## 3.3. Baselines and Special Cases

**Maximum likelihood generative modeling special case:** We start with the following special case, where the learning problem simplifies to maximum likelihood generative modeling with the log-likelihood objective.

**Lemma 3.4.** *Let the given observations equal the ground-truth states, i.e., $X_i = S_i^*$ such that $\Pr(X_i \mid S) = 1$ if $S = X_i = S_i^*$ and 0 otherwise. Then, Problem 2.1 of solving $argmax_\theta \Pr(X_{1:N} \mid \theta)$ simplifies to $argmax_\theta \sum_{i=1}^N \log(\Pr(S_i^* \mid \theta))$.*

Many approaches exist for solving this problem, such as variational autoencoders (VAE) (Kingma & Welling, 2013), discrete diffusion (Austin et al., 2021), and autoregressive generation (Bengio et al., 2003). Furthermore, if each $X_i = S_i^*$ uniquely determines one trajectory $\tau$, then GReinSS simplifies to autoregressive generation as shown in Theorem A.2

**Local search baseline:** We obtain an additional special case if we know that each indirect observation $X_i$ can be explained by exactly one latent state $S \in \mathcal{S}$.

**Observation 3.5.** *If for an observation $X_i$ there is exactly one latent state $\hat{S}_i \in \mathcal{S}$ such that $\Pr(X_i \mid \hat{S}_i)$ is non-zero then $argmax_S \Pr(X_i \mid S)\Pr(S \mid \theta) = argmax_S \Pr(X_i \mid S) = \hat{S}_i$ for any $\theta$.*

Thus, if for each $X_i$ the probability $\Pr(X_i \mid S)$ is non-zero for only one state $S$, then Problem 2.2 simplifies to $\hat{S}_i = \arg\max_S \Pr(X_i \mid S)$. To provide intuition, we implement an extremely simple local search baseline that directly optimizes $\arg\max_S \Pr(X_i \mid S)$ without using any model $\Pr(S \mid \theta)$ with details in Section B.8. This baseline can be effective in cases where $\Pr(X_i \mid S)$ is near zero for all but one state $S$ (as shown in Section 4.1).

**Expectation maximization baselines:** Expectation maximization consists of alternating between an E-step of estimating latent states $\hat{S}_{1:N}$ and an M-step of optimizing the parameters $\theta$. Specifically, we define $q(S \mid X_{1:N}, \theta) = \frac{1}{N} \sum_{i=1}^N \Pr(X_i \mid S) \Pr(S \mid \theta)$. Intuitively, this distribution is approximately an averaged probability distribution over predicted states $\hat{S}_i$ for $i \in [N]$, since the optimal predicted state $\hat{S}_i = \arg\max_S \Pr(X_i \mid S) \Pr(S \mid \theta)$. As shown in Theorem A.3, expectation maximization alternates between an E-step of estimating latent states $S$ (via the term $q(S \mid X_{1:N}, \theta)$) and an M-step of maximizing the complete-data log-likelihood expectation value using $\theta^{(t+1)} = \arg\max_\theta \mathbb{E}_{S \sim q(S \mid X_{1:N}, \theta^{(t)})}[\log(\Pr(S \mid \theta))]$. The term being maximized only involves $S$ and $\theta$ but not $X_{1:N}$ due to the fact that $X_{1:N}$ only depends on $\theta$ through $S$ (Theorem A.3). Consequently, the M-step is simply selecting $\theta$ to maximize the likelihood of a known distribution of states, similarly to solving Problem 2.1 in the special case where $S_{1:N}^*$ is known (as in Observation 3.4).

For generalized expectation maximization (GEM), the M-step consists of updating $\theta$ to increase the log-likelihood of the distribution of states rather than solving for the optimal $\theta$. Exactly solving GEM cannot be utilized as a baseline method since the E-step cannot be exactly computed for problems with an exponentially large space $\mathcal{S}$ of latent states. In principle, the E-step could also be approximated using variational inference by introducing an auxiliary posterior inference model; however, knowing $\Pr(X \mid S)$ enables the simpler approximation of using the current $\theta$ to identify latent states $\hat{S}_i$ that maximize $\Pr(X_i \mid S) \Pr(S \mid \theta)$ as approximations of $S_{1:N}^*$ (i.e. an instance of Problem 2.2). These inferred latent states $\hat{S}_{1:N}$ are given equal probability, and other states are given probability 0 rather than marginalizing over the entire exponentially large state space $\mathcal{S}$. Consequently, our approximation of GEM consists of alternating between an approximated E-step of inferring $\hat{S}_{1:N}$ and an exact M-step of updating $\theta$ to increase the log-likelihood $\sum_{i=1}^N \log(\Pr(S_i^* \mid \theta))$ via gradient descent. Additional details and methodological comparisons with GReinSS are provided in Section B.3.

To serve as baselines, this algorithm can easily be combined with standard methods for maximum likelihood generative modeling. We therefore solve GEM using variational autoencoders, discrete diffusion, and autoregres-sive models as shown in Table 1 (details provided in Sections B.5, B.7 and B.8). Problem 2.2 is solved by sampling $S$ from $\Pr(S \mid \theta)$ and then predicting states that maximize $\Pr(S \mid \theta) \Pr(X_i \mid S)$ (details in Section B.5).

**Policy learning baselines:** In one special case, GReinSS simplifies to standard policy gradients.

**Lemma 3.6.** *Let $\Pr(X_i \mid S) = \Pr(X_j \mid S)$ across all $S \in \mathcal{S}$ for all $i, j \in [N]$. Then, GReinSS simplifies to standard policy gradients with rewards $r'(\tau) = \sum_{i=1}^N \Pr(X_i \mid \tau)$ and standard reward normalization.*

Intuitively, the denominator $\Pr(X_i \mid \theta)$ in the GReinSS reward function (3) helps ensure the trajectory probabilities are balanced across $X_{1:N}$ rather than being overly narrow (as illustrated in Section B.2 and Figure S1). However, when $\Pr(X_i \mid \theta)$ is the same for all $i$, this term becomes irrelevant and the reward can simplify to $\sum_{i=1}^N \Pr(X_i \mid \tau)$. We define *naive policy gradients* as a simple ablation of GReinSS, where we use the reward function $r'(\tau) = \sum_{i=1}^N \Pr(X_i \mid \tau)$, removing the denominator $\Pr(X_i \mid \theta)$. As an alternative baseline, one can better balance probabilities between trajectories without the denominator term $\Pr(X_i \mid \theta)$ by using trajectory balance GFlowNets (Malkin et al., 2022). This allows the probability to be balanced across trajectories $\tau$, supporting multiple observations $X_i$ while still using the predefined reward function $r'(\tau) = \sum_{i=1}^N \Pr(X_i \mid \tau)$. GFlowNets solve Problem 2.1 optimally in the following special case.

**Lemma 3.7.** *Let, for each $i \in [N]$, $\Pr(X_i \mid \tau) = 1$ for exactly one trajectory $\tau$ and 0 for all other trajectories. Then, the optimal GFlowNets distribution $\Pr(\tau \mid \theta)$ is also the optimal solution to Problem 2.1.*

For this case, Problem 2.2 may be solved by sampling $\tau$ from $\Pr(\tau \mid \theta)$ and then predicting states $S = S(\tau)$ that maximize $\Pr(S \mid \theta) \Pr(X_i \mid S)$ (see Section B.5).

# 4. Results

## 4.1. Simulations

We compare GReinSS to the baseline methods listed in Table 1 on two simulation experiments, where the latent states correspond to distinct combinatorial objects, namely graphs and sets. In the first experiment, our latent states are directed graphs representing some hidden process, and our observations are lists of start and end points of random walks in these directed graphs (Figure 2a).

**Problem 4.1** (PROCESS GRAPH INFERENCE FROM RANDOM WALK ENDPOINTS). Find the directed graphs $S_{1:N}^*$ drawn from an unknown distribution $\Pr^*(S)$ given observations $X_{1:N}$ with each $X_i$ composed of $k$ pairs of start and end vertices of $k$ absorbing random walks from $S_i^*$.

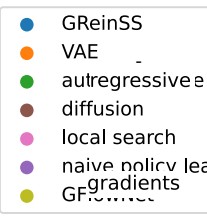

- ● GReinSS
- ● VAE
- ● autregressive a
- ● diffusion
- ● local search
- ● naive policy lea
  gradients
- ● GFlowNet

*Figure 2.* **GReinSS outperforms all baseline methods in simulated process graph inference. a** An example latent graph $S_i^*$ and observation $X_i$ consisting of the start and end points of $k$ random walks. **b** $F_1$-score as a function of the number $k$ of random walks.

The absorbing random walks are unbiased random walks through the graph $S_i^*$ with a single terminating self-loop on each node, as done in (Wu et al., 2012). This very general problem could represent inferring transportation networks (Biagioni & Eriksson, 2012), biochemical process graphs (Friedman et al., 2000), information transmission graphs (Gomez-Rodriguez et al., 2012), etc. To apply the GReinSS framework, we must recast the above problem by (i) parameterizing a procedure for generating latent state graphs, and (ii) specifying the probability $\Pr(X_i \mid S)$ of generating our indirect observations. Rather than assuming any knowledge on the distribution $\Pr^*(S)$ of graphs, we define $\theta$ as the parameters of a neural network that generates graphs $S$ by sequentially adding directed edges to an initially empty graph, ending with a termination action (model details provided in Section B.7).

The probability $\Pr(X_i \mid S)$ is the product of the start and end probabilities that follow from the inverse shifted Laplacian matrix $(L + I)^{-1}$ as described in (Wu et al., 2012). As such, we can solve Problem 4.1 naturally using GReinSS by splitting it into a learning problem of identifying parameters $\theta$ generating the distribution of graphs $\Pr(S \mid \theta)$ (Problem 2.1), and an inference problem of estimating each $S_i^*$ given $X_i$ and $\theta$ (Problem 2.2). We generate three simulation instances each composed of $N = 1000$ observations but a varying number $k \in \{10, 100, 1000\}$ of random walks. To do so, we begin by generating a base graph via Erdős–Rényi with an edge inclusion probability $1/2$ (Erdős & Rényi, 1960). Next, we assign a weight to each edge sampled from $U(1/4, 1)$ (with the lower limit set to $1/4$ to ensure edge recurrence across latent graphs $S_i^*$). We apply weight thresholding with thresholds sampled from $U(0, 1)$ to generate 1000 latent states $S_{1:1000}^*$. Finally, we generate $k$ random walks recording the pair $(v, w)$ of vertices corresponding to a start node $v$ sampled uniformly at random and the ending node $w$ obtained via an unbiased random walk process (Section C.1).

We measure graph reconstruction accuracy as an $F_1$ score,

i.e. the harmonic mean of precision and recall of directed edge sets (Section C.2). As shown in Figure 2b and Figure S3, we find that GReinSS consistently achieves the highest median $F_1$ score, followed by the GEM-based baselines using VAE and autoregression. These latter two methods perform very similarly, indicating they may both be effectively solving the GEM M-step, with their performance fundamentally limited by the GEM framework. With the exception of GFlowNets and naive policy gradients, the $F_1$ scores consistently increase with the number $k$ of random walks, showing most approaches can be effective with sufficiently informative observations. Strikingly, for $k = 10$, GReinSS has the highest median $F_1$ score of 0.891, with all baselines having median $F_1$ scores below 0.55 due to the small amount of information per observation. Despite naive policy gradients and GFlowNets being the most structurally similar baseline methods to GReinSS, they perform very poorly with median $F_1$ scores consistently below 0.55. Specifically, Naive Policy Gradient consistently predicted an empty graph, achieving a high reward for a small number of observations but an $F_1$ score of 0. This demonstrates that while the dynamic reward function is a relatively small algorithmic change to policy gradients, it is absolutely essential for achieving strong results.

In the second experiment, our latent states are a family $S_{1:N}^*$ of $N = 100$ subsets from a universe $\mathcal{U}$, and observations $X_{1:N}$ are noisy real-valued vectors indicating which elements are more likely to be present in each set (Figure 3a).

**Problem 4.2** (SUBSET INFERENCE FROM NOISY ELEMENT MEASUREMENTS). Find the family $S_{1:N}^*$ of subsets drawn from an unknown distribution $\Pr^*(S)$ on a fixed universe $\mathcal{U}$, given observations $X_{1:N}$ each composed of a vector of length $|\mathcal{U}|$ with $X_{i,j}$ drawn from a known distribution $D^+$ if $j \in S_i^*$ and $X_{i,j}$ drawn from $D^-$ otherwise.

The above problem where noisy observations indicate which conditions are more likely true for each data point arises in many scientific contexts, including detecting chemicals from noisy mass spectrometry intensities (Aebersold & Mann, 2003), determining active genes from noisy expression data (Schena et al., 1995), etc. This problem fits naturally within the GReinSS framework: Latent states $S$ are generated by adding elements from $\mathcal{U}$ iteratively, starting with the empty set $\emptyset$, ending with a termination action using a neural network with parameters $\theta$ described in Section B.7. The probability distribution $\Pr(X_i \mid S)$ is derived trivially from the given distributions $D^+$ and $D^-$ as $\prod_{j \in S} D^+(X_{i,j}) \prod_{j \in \mathcal{U} \setminus S} D^-(X_{i,j})$. As with Problem 4.1, we solve Problem 4.2 in a two-step fashion by applying GReinSS to first solve Problem 2.1 followed by Problem 2.2.

We generate simulation instances with varying noise levels $\sigma \in \{0.1, 0.2, 0.3, 0.4, 0.5\}$ and varying universe sizes $|\mathcal{U}| \in \{10, 100, 1000\}$. Specifically, we model the distribu-

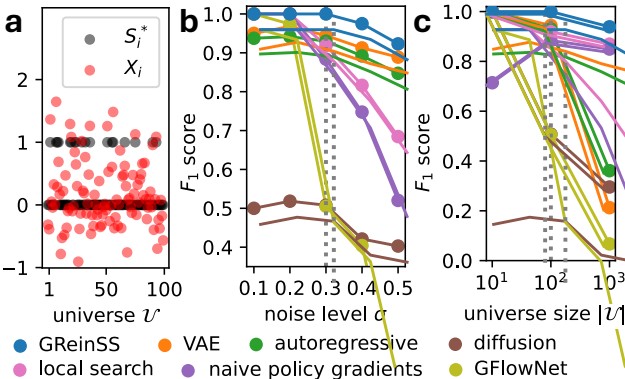

*Figure 3.* **GReinSS outperforms all baseline methods in simulated subset inference. a** An example set $S_i^*$ and noisy observation $X_i$. **b** GReinSS outperforms baselines for all noise levels $\sigma$. **c** GReinSS scales to large universes $\mathcal{U}$ unlike baseline methods.

tions $D^+$ and $D^-$ as Gaussian distributions $N(1, \sigma^2)$ and $N(0, \sigma^2)$ using the specified noise level $\sigma$ as the standard deviation. We simulate the latent states $S_i^*$ using a process inspired by dictionary learning (Aharon et al., 2006), in which each set $S_i^*$ is the union of random reusable module subsets (Mairal et al., 2009) with details provided in Section C.3. The state-generating process $\Pr^*(S)$ results in states that exhibit shared structural patterns while maintaining a high degree of random variation. The combination of a widely dispersed and highly randomized state-generation process $\Pr^*(S)$ and very informative observations $X_i$ (especially for small $\sigma$) makes off-policy learning extremely valuable. While it is hard to sample exactly from the optimal off-policy sampling distribution $\Pr(S \mid X_{1:N}, \theta)$, one can bias the sampling $\Pr(S \mid \theta)$ towards $\Pr(S \mid X_{1:N}, \theta)$ using the observations $X_{1:N}$ as described in Section C.4. We use off-policy learning for all policy learning methods, including GReinSS, naive policy gradients, and GFlowNets. As shown in Figure S4, the accuracy of GReinSS is not sensitive to the exact off-policy sampling proposal so long as it biases sampling toward $\Pr(S \mid X_{1:N}, \theta)$.

We assess performance using $F_1$ scores, comparing ground-truth subsets $S_i^*$ to predicted subsets $\hat{S}_i$. For simulation instances with a universe with $|\mathcal{U}| = 100$ elements, we find that GReinSS consistently outperforms the baseline methods across varying noise levels (Figure 3b and Figure S6). For low $\sigma < 0.3$, the best alternatives are naive policy gradients, local search, and GFlowNets, whereas for high $\sigma > 0.3$, the best alternatives are VAE and autoregressive. This illustrates how for low $\sigma$ the key factor is utilizing the observations either via off-policy learning or local search, whereas for high $\sigma$ the key factor is effectively optimizing $\Pr(X_{1:N} \mid \theta)$ using either GReinSS or GEM. Additionally, we analyze the set reconstruction $F_1$ score while varying the size $|\mathcal{U}|$ of the universe but keeping $\sigma = 0.3$ (Figure 3c and Figure S5). For tiny vectors of $|\mathcal{U}| = 10$ elements, all methods other than naive policy gradients achieve a perfect

median $F_1$ score of 1.0. However, all baseline methods other than local search and naive policy gradients scale catastrophically to universe sizes of $|\mathcal{U}| = 1000$ (median $F_1 < 0.4$). Meanwhile, for universe sizes of $|\mathcal{U}| = 1000$, GReinSS achieves a median $F_1$ score of 0.938, while local search and naive policy gradients achieve median $F_1$ scores of 0.869 and 0.849, respectively. This demonstrates that simple approaches are sufficient for very small universe sizes, but only GReinSS effectively scales to large vectors. In Figure S2, we analyze policy learning methods without off-policy learning, showing that GReinSS still performs best.

## 4.2. RNA Splicing from Short Read RNA-seq Data

Cells produce proteins by transcribing DNA into precursor RNA, splicing it into a mature RNA transcript, and translating it into a protein (Alberts et al., 1994). While transcription and translation are mostly deterministic, splicing is a highly variable process that selects and connects contiguous nucleotide segments, called *exons*. As such, this process yields multiple distinct RNA molecules, called *isoforms*, from the same gene (Wang et al., 2008) as shown in Figure 4a. Long-read RNA sequencing directly observes full-length isoforms but is expensive (Tilgner et al., 2015; Au et al., 2013). In contrast, inexpensive short-read RNA sequencing produces short (around 100 nucleotide) reads from mature transcripts. As such, isoforms and their proportions must be inferred indirectly from these reads (Mortazavi et al., 2008; Trapnell et al., 2010; Li & Dewey, 2011). Here, we propose to use GReinSS for this problem as follows.

**Problem 4.3** (ISOFORM PROPORTION INFERENCE FROM READ COUNTS). Given a set $X_{1:N}$ of reads aligned to one gene across $M$ sequencing samples, with each read $X_i$ consisting of a sample index and genomic position, estimate the distribution $\Pr^*(S)$ of isoforms in each of the $M$ samples.

To apply the GReinSS framework, we first (i) parameterize a procedure for generating latent state isoforms and (ii) specify the probability $\Pr(X_i \mid S)$ of generating our indirect observations. We cast each latent state $S$ as a tuple containing the genomic regions (exons) of the isoform, the sample index of the isoform, and the genomic position of the read $X_i$ produced from this isoform (Figure 4b). The isoform is generated using a neural network with parameters $\theta$ by iteratively selecting transitions (junctions) between included contiguous genomic regions (exons), constructing an isoform as described in Section C.5. Our formulation of latent states implies $\Pr(X_i \mid S) = 1$ if the genomic position and sample index of $S$ and $X_i$ agree, and 0 otherwise.

To evaluate GReinSS on this problem, we utilize the GTEx database (Lonsdale et al., 2013), containing 17,371 human tissue samples with short-read sequencing data, of which 61 also have matched long-read sequencing data. Specifi-

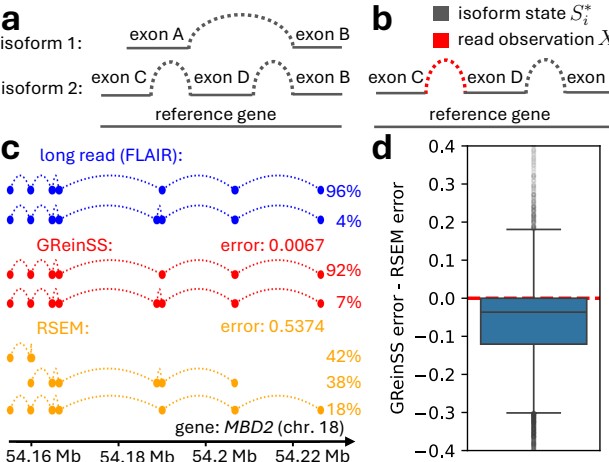

*Figure 4.* **GReinSS outperforms RSEM in predicting RNA isoforms from short-read sequencing data. a** A diagram of two distinct isoforms of one gene, with solid lines for exons and dashed lines for junctions (transitions between exons). **b** A diagram of an observation read $X_i$ covering some junction in the isoform $S_i^*$. **c** The isoforms detected by long-read sequencing, as well as those predicted by GReinSS and RSEM, are shown for an example gene (*MBD2*) in one cultured fibroblast sample. Dots show exons and dashed lines show (much longer) junctions. Unlike RSEM, GReinSS reconstructs the same two isoforms identified in the long-read data with very similar proportions. **d** The GReinSS error minus the RSEM error is shown for all 14,390 genes, with a median difference in errors of $-0.0405$.

cally, GTEx contains: (i) short-read sequencing junction-overlapping read counts calculated using the STAR (Dobin et al., 2013) aligner; (ii) isoforms and their proportions estimated from short-read sequencing data using RSEM (Li & Dewey, 2011); and (iii) isoforms and their proportions estimated from long-read sequencing data calculated using FLAIR (Tang et al., 2020). We use (i) as input to GReinSS, (ii) as a baseline method, and (iii) as ground-truth for evaluation. Note that the baseline method RSEM is a commonly used expectation maximization-based algorithm for isoform quantification, which, similarly to GReinSS, uses the same splice-aware read alignments as input. We also include comparisons with the naive policy gradients and GFlowNets ablations in Figure S8, showing that these ablations underperformed both GReinSS and RSEM. We focus our analysis on the $M = 61$ tissue samples with matched long-read data. Moreover, we restrict our analysis to a total of 14,390 human genes that had a sufficient number ($\geq 100$) of junction-overlapping reads in the short-read data as well as isoform-covering reads in the long-read data.

To evaluate prediction accuracy, we must account for (i) errors in the exons included in each isoform, (ii) errors in predicted proportions, and (iii) the varying long-read sequencing support across samples. We achieve this by utilizing (i) pairwise Jaccard distance between loci covered by

isoforms, (ii) optimal transport across isoform proportions, and (iii) a sample average weighted by long-read support (details and an example provided in Section C.6 and Figure S7). Briefly, the *isoform prediction error* is 0 if correct isoforms and proportions are predicted across all samples and 1 if all predicted isoforms have zero overlap with all long-read-supported isoforms.

Figure 4c shows the isoform reconstruction for the gene *MBD2* in one cultured fibroblast sample ("GTEX-QV44-0008-SM-447AX"). The ground-truth long-read sequencing detected one isoform with 96% proportion and one isoform with 4% proportion, which we will refer to as the major and minor isoform, respectively. These isoforms are similar with a Jaccard distance of 0.1495. For GReinSS, the major isoform has 92% proportion and the minor isoform has 7% proportion (with other isoforms totaling < 1%). Applying optimal transport gives an error of 0.0067 due to a slight proportion inaccuracy on the two highly similar correct isoforms. For RSEM, the major isoform has 18% proportion, with other highly distinct isoforms totaling 82% proportion, resulting in an error of 0.5374. Taking an average across 61 samples weighted by the long-read sequencing counts gives an error of 0.0069 for GReinSS and 0.471 for RSEM, leading to a difference of $0.0069 - 0.471 = -0.4641$, showing that the isoforms identified by GReinSS better match the long-read ground truth. Looking across all genes, Figure 4d and Figure S9 show that GReinSS outperforms RSEM much more frequently than vice versa. Specifically, on 46.6% of genes GReinSS outperforms RSEM by at least 0.05, whereas only on 9.4% of genes RSEM outperforms GReinSS by at least 0.05. This demonstrates how a straightforward application of GReinSS can improve upon the techniques being used in major real-world applications.

## 5. Conclusion

In this work, we introduced GReinSS, a novel framework for learning distributions over discrete latent states from indirect observation data. GReinSS directly optimizes the observation data likelihood without utilizing ground-truth latent states. To achieve this, GReinSS dynamically rescales rewards so that the policy gradient is an unbiased estimator of the observation data log-likelihood gradient. Consequently, GReinSS allows policy learning to be an effective alternative to expectation maximization for combinatorially large latent spaces where an exact implementation of expectation maximization is infeasible.

On simulated latent graph and latent set inference problems, GReinSS consistently outperforms baseline methods, including (naive) policy gradients, GFlowNets, local search, and GEM-based implementations of VAEs, autoregressive models, and discrete diffusion. In particular, the poor performance of the naive policy gradients ablation

of GReinSS shows the vital importance of GReinSS's dynamic reward function. On the biological task of isoform inference from short-read RNA sequencing data, GReinSS outperforms the standard EM-based algorithm RSEM used in the GTEx database. This illustrates the real-world effectiveness of GReinSS on practical problems beyond prior works (Ivanovic & El-Kebir, 2023; 2025).

Beyond applying the GReinSS framework to other real-world problems, there is great potential for methodological extensions. First, GReinSS currently uses policy gradients, but could be extended to use Q learning (Watkins & Dayan, 1992) or the actor-critic framework (Sutton et al., 1999). Second, although we derived the optimal off-policy sampling distribution (Theorem 3.3), this and prior work rely on heuristically defined off-policy sampling (Ivanovic & El-Kebir, 2023; 2025). Alternatively, one could train an auxiliary network to learn an off-policy sampling distribution that approximates the optimal distribution $q(\tau \mid X_{1:N}, \theta)$. Third, an approximate probability function $\Pr(X \mid S)$ of observations $X$ given some state $S$ is currently assumed to be known but could instead contain learned parameters to account for unknown aspects of observation generation. Fourth, more sophisticated forms of multi-environmental modeling could be implemented. The isoform inference application utilizes a fairly simple multi-environment setup, but future extensions could utilize more complex differences in the distributions $\Pr(S \mid \theta)$ and $\Pr(X_i \mid S)$ across environments, including the addition of individualized environment-specific parameters. Overall, this work provides a starting point for a wide variety of future applications and extensions.

## Acknowledgments

This research was supported by National Science Foundation grant CCF-2046488, the Molecule Maker Lab Institute: An AI Research Institutes program supported by NSF under Award No. 2505932, and the DOE Center for Advanced Bioenergy and Bioproducts Innovation (U.S. Department of Energy, Office of Science, Biological and Environmental Research Program under Award Number DE-SC0018420). Any opinions, findings, and conclusions or recommendations expressed in this publication are those of the author(s) and do not necessarily reflect the views of the U.S. Department of Energy.

## Impact Statement

This paper introduces new machine learning methodologies with the primary goal of advancing the field of machine learning. This work is primarily applicable to problems involving latent-variable inference from indirect observations, including scientific and biological data analysis. Potential downstream impacts of this work may include improved analysis of biological systems or other forms of experimental data. We do not anticipate that this work introduces novel ethical concerns beyond those broadly applicable to the development of novel machine learning techniques, and we do not identify any specific societal impacts requiring special consideration.

## Code Availability

Our implementation of GReinSS is available at `https://github.com/elkebir-group/GReinSS`.

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

# A. Proofs

**(Main Text) Theorem 3.1.** The policy gradient $\mathbb{E}_\tau[r(\tau)\frac{d}{d\theta}\log(\Pr(\tau \mid \theta))]$ with dynamically changing rewards

$$r(\tau) = \sum_{i=1}^{N} \frac{\Pr(X_i \mid \tau)}{\Pr(X_i \mid \theta)}. \tag{4}$$

is an unbiased estimator of the gradient $\frac{d}{d\theta}\log(\Pr(X_{1:N} \mid \theta))$ of the log-likelihood objective. That is,

$$\frac{d}{d\theta}\log(\Pr(X_{1:N} \mid \theta)) = \mathbb{E}_\tau[r(\tau)\frac{d}{d\theta}\log(\Pr(\tau \mid \theta))]. \tag{5}$$

*Proof.* Let $\mathcal{T}$ be the set of all trajectories. We then have the following.

$$\frac{d}{d\theta}\log(\Pr(X_1,\ldots X_N \mid \theta)) = \frac{d}{d\theta}\sum_{i=1}^{N}\log(\sum_{\tau\in\mathcal{T}}\Pr(X_i \mid \tau)\Pr(\tau \mid \theta)) = \sum_{i=1}^{N}\frac{\sum_{\tau\in\mathcal{T}}\Pr(X_i \mid \tau)\frac{d}{d\theta}\Pr(\tau \mid \theta)}{\sum_{\tau\in\mathcal{T}}\Pr(X_i \mid \tau)\Pr(\tau \mid \theta)} \tag{6}$$

$$= \sum_{\tau\in\mathcal{T}}\sum_{i=1}^{N}\frac{\Pr(X_i \mid \tau)}{\Pr(X_i \mid \theta)}\frac{d}{d\theta}\Pr(\tau \mid \theta) = \sum_{\tau\in\mathcal{T}}\sum_{i=1}^{N}\frac{\Pr(X_i \mid \tau)}{\Pr(X_i \mid \theta)}\Pr(\tau \mid \theta)\frac{d}{d\theta}\log(\Pr(\tau \mid \theta)) \tag{7}$$

$$= \sum_{\tau\in\mathcal{T}}\Pr(\tau \mid \theta)\left(\sum_{i=1}^{N}\frac{\Pr(X_i \mid \tau)}{\Pr(X_i \mid \theta)}\right)\frac{d}{d\theta}\log(\Pr(\tau \mid \theta)) = \mathbb{E}_\tau[\left(\sum_{i=1}^{N}\frac{\Pr(X_i \mid \tau)}{\Pr(X_i \mid \theta)}\right)\frac{d}{d\theta}\log(\Pr(\tau \mid \theta))] \tag{8}$$

$$= \mathbb{E}_\tau[r(\tau)\frac{d}{d\theta}\log(\Pr(\tau \mid \theta))] \tag{9}$$

$\square$

**(Main Text) Theorem 3.3.** The unbiased variance-minimizing off-policy sampling proposal is $q(\tau \mid X_{1:N}, \theta) = \frac{1}{N}\sum_{i=1}^{N}\Pr(\tau \mid X_i, \theta)$.

*Proof.* Existing work (Kahn & Marshall, 1953; Owen, 2013) has proven the unbiased variance-minimizing sampling proposal is $\Pr(\tau) \propto |r(\tau)|\Pr(\tau \mid \theta)$. We then have the following.

$$|r(\tau)|\Pr(\tau \mid \theta) = \sum_{i=1}^{N}\frac{\Pr(X_i \mid \tau)}{\Pr(X_i \mid \theta)}\Pr(\tau \mid \theta) = \sum_{i=1}^{N}\Pr(\tau \mid X_i, \theta) = Nq(\tau \mid X_{1:N}, \theta). \tag{10}$$

Dividing this term by $N$ normalizes it to give a total probability of 1 across all trajectories. Thus, the optimal off-policy sampling proposal is $q(\tau \mid X_{1:N}, \theta)$. $\square$

**(Main Text) Lemma 3.4.** Let the given observations equal the ground-truth states, i.e., $X_i = S_i^*$ such that $\Pr(X_i \mid S) = 1$ if $S = X_i = S_i^*$ and 0 otherwise. Then, Problem 2.1 of solving $\text{argmax}_\theta \Pr(X_{1:N} \mid \theta)$ simplifies to $\text{argmax}_\theta \sum_{i=1}^{N}\log(\Pr(S_i^* \mid \theta))$.

*Proof.*

$$\text{argmax}_\theta \Pr(X_{1:N} \mid \theta) = \text{argmax}_\theta \log(\Pr(X_{1:N} \mid \theta)) \tag{11}$$

$$= \text{argmax}_\theta \log(\prod_{i=1}^{N}\Pr(X_i \mid \theta)) \tag{12}$$

$$= \text{argmax}_\theta \sum_{i=1}^{N}\log(\Pr(X_i \mid \theta))) \tag{13}$$

$$= \text{argmax}_\theta \sum_{i=1}^{N}\log(\Pr(S_i^* \mid \theta))) \tag{14}$$

$\square$

**(Main Text) Lemma 3.6.** Let $\Pr(X_i \mid S) = \Pr(X_j \mid S)$ across all $S \in \mathcal{S}$ for all $i, j \in [N]$. Then, GReinSS simplifies to standard policy gradients with rewards $r'(\tau) = \sum_{i=1}^{N} \Pr(X_i \mid \tau)$ and standard reward normalization.

*Proof.* GReinSS utilizes the reward function $r(\tau) = \sum_{i=1}^{N} \Pr(X_i \mid \tau) / \Pr(X_i \mid \theta)$. The naive policy learning reward function is $r'(\tau) = \sum_{i=1}^{N} \Pr(X_i \mid \tau)$. Standard reward normalization modifies this to $r'_N(\tau) = r'(\tau) / \mathbb{E}_{\tau' \sim \Pr(\tau' \mid \theta)}[r'(\tau')]$. Note that for all $i, j \in [N]$ we have $\Pr(X_i \mid \theta) = \Pr(X_j \mid \theta)$ since $\Pr(X_i \mid S) = \Pr(X_j \mid S)$ for all $S \in \mathcal{S}$. Also note that for all $i, j \in [N]$ and all trajectories $\tau$, we have $\Pr(X_i \mid \tau) = \Pr(X_i \mid S(\tau)) = \Pr(X_j \mid S(\tau)) = \Pr(X_j \mid \tau)$. We then have the following.

$$r'_N(\tau) = \frac{r'(\tau)}{\mathbb{E}[r'(\tau)]} \tag{15}$$

$$= \frac{\sum_{i=1}^{N} \Pr(X_i \mid \tau)}{\mathbb{E}_{\tau' \sim \Pr(\tau' \mid \theta)}[\sum_{i=1}^{N} \Pr(X_i \mid \tau)]} \tag{16}$$

$$= \frac{N \Pr(X_1 \mid \tau)}{\mathbb{E}_{\tau' \sim \Pr(\tau' \mid \theta)}[N \Pr(X_1 \mid \tau)]} \tag{17}$$

$$= \frac{\Pr(X_1 \mid \tau)}{\mathbb{E}_{\tau' \sim \Pr(\tau' \mid \theta)}[\Pr(X_1 \mid \tau')]} \tag{18}$$

$$= \frac{\Pr(X_1 \mid \tau)}{\Pr(X_1 \mid \theta)} \tag{19}$$

$$= \frac{1}{N} \sum_{i=1}^{N} \frac{\Pr(X_i \mid \tau)}{\Pr(X_i \mid \theta)} \tag{20}$$

$$= \frac{1}{N} r(\tau). \tag{21}$$

Thus, $r'_N(\tau)$ differs from $r(\tau)$ only by a constant, making them equivalent to optimize. $\square$

**(Main Text) Lemma 3.7.** Let, for each $i \in [N]$, $\Pr(X_i \mid \tau) = 1$ for exactly one trajectory $\tau$ and 0 for all other trajectories. Then, the optimal GFlowNets distribution $\Pr(\tau \mid \theta)$ is also the optimal solution to Problem 2.1.

*Proof.* For each $i \in [N]$ let $\tau_i$ be the trajectory for which $\Pr(X_i \mid \tau) = 1$. Let $\mathbb{I}_{\tau = \tau_i}$ equal 1 if $\tau = \tau_i$ and 0 otherwise. The GFlowNets reward function is $r'(\tau) = \sum_{i=1}^{N} \Pr(X_i \mid \tau) = \sum_{i=1}^{N} \mathbb{I}_{\tau = \tau_i}$. The optimal solution for GFlowNets is having the probability $\Pr(\tau \mid \theta)$ be proportional to the reward $r'(\tau)$. Consequently, the optimal solution for GFlowNets is achieved by setting $\Pr(\tau_i \mid \theta) = \frac{1}{N} \sum_{i=1}^{N} \mathbb{I}_{\tau = \tau_i}$. For GReinSS we have the following.

$$\operatorname{argmax}_\theta \Pr(X_1, \ldots X_N \mid \theta) = \operatorname{argmax}_\theta \sum_{i=1}^{N} \log(\mathbb{E}_{\tau \sim \Pr(\tau \mid \theta)}[\Pr(X_i \mid \tau)]) \tag{22}$$

$$= \operatorname{argmax}_\theta \sum_{i=1}^{N} \log(\mathbb{E}_{\tau \sim \Pr(\tau \mid \theta)}[\mathbb{I}_{\tau = \tau_i}]) \tag{23}$$

$$= \operatorname{argmax}_\theta \sum_{i=1}^{N} \log(\Pr(\tau_i \mid \theta)). \tag{24}$$

This log-likelihood is maximized by setting $\Pr(\tau_i \mid \theta)$ to the empirical distribution in the data $\mathbb{E}_{i \in [N]} \mathbb{I}_{\tau = \tau_i}$. Thus, the optimal solutions for GReinSS and GFlowNets in this special case are both $\Pr(\tau_i \mid \theta) = \frac{1}{N} \sum_{i=1}^{N} \mathbb{I}_{\tau = \tau_i}$.

$\square$

Mini-batching is compatible with GReinSS and still gives an unbiased estimator of the gradient of the observation data log-likelihood as shown below.

**Corollary A.1.** *Let $B \subset [N]$ be a mini-batch sampled uniformly at random, and define the mini-batch reward*

$$r_B(\tau) = \sum_{i \in B} \frac{\Pr(X_i \mid \tau)}{\Pr(X_i \mid \theta)}. \tag{25}$$

*Then the policy gradient*

$$\frac{N}{|B|} \mathbb{E}_{\tau, B}[r_B(\tau) \frac{d}{d\theta} \log \Pr(\tau \mid \theta)] \tag{26}$$

*is an unbiased estimator of the full-data log-likelihood gradient*

$$\frac{d}{d\theta} \log \Pr(X_{1:N} \mid \theta). \tag{27}$$

*Proof.*

$$\frac{N}{|B|} \mathbb{E}_{\tau, B}[r_B(\tau) \frac{d}{d\theta} \log \Pr(\tau \mid \theta)] = \mathbb{E}_{\tau, B}[\frac{N}{|B|} \sum_{i \in B} \frac{\Pr(X_i \mid \tau)}{\Pr(X_i \mid \theta)} \frac{d}{d\theta} \log \Pr(\tau \mid \theta)] \tag{28}$$

$$= \mathbb{E}_{\tau}[\mathbb{E}_B[\frac{N}{|B|} \sum_{i \in B} \frac{\Pr(X_i \mid \tau)}{\Pr(X_i \mid \theta)}] \frac{d}{d\theta} \log \Pr(\tau \mid \theta)] = \mathbb{E}_{\tau}[\sum_{i=1}^{N} \frac{\Pr(X_i \mid \tau)}{\Pr(X_i \mid \theta)} \frac{d}{d\theta} \log \Pr(\tau \mid \theta)] \tag{29}$$

This is then equal to $\frac{d}{d\theta} \log \Pr(X_{1:N} \mid \theta)$ by Theorem 3.1.

$\square$

If each $X_i = S_i^*$ uniquely determines one trajectory $\tau$, then GReinSS simplifies to autoregressive generation as shown below.

**Lemma A.2.** *Autoregressive generation using the negative log-likelihood loss is equivalent to GReinSS when for all $i \in [N]$ the probability $\Pr(X_i \mid \tau)$ is non-zero for exactly one known trajectory $\tau_i^*$.*

*Proof.* For $i \in [N]$, let $\tau_i^*$ be the one trajectory for which $\Pr(X_i, \tau_i^*)$ is non-zero. Define $\delta(\tau, \tau') = 1$ if $\tau = \tau'$ and $0$ otherwise. The off-policy sampling has the following probability.

$$\Pr(\tau \mid X_1, \ldots, X_N, \theta) = \frac{1}{N} \sum_{i=1}^{N} \Pr(\tau \mid X_i, \theta) \tag{30}$$

$$= \frac{1}{N} \sum_{i=1}^{N} \frac{\Pr(\tau \mid \theta) \Pr(X_i \mid \tau)}{\Pr(X_i \mid \theta)} \tag{31}$$

$$= \frac{1}{N} \sum_{i=1}^{N} \frac{\Pr(\tau \mid \theta) \Pr(X_i \mid \tau)}{\sum_{\tau' \in \mathcal{T}} \Pr(X_i \mid \tau') \Pr(\tau' \mid \theta)} \tag{32}$$

$$= \frac{1}{N} \sum_{i=1}^{N} \frac{\Pr(\tau_i^* \mid \theta) \Pr(X_i \mid \tau_i^*) \delta(\tau, \tau_i^*)}{\Pr(X_i \mid \tau_i^*) \Pr(\tau_i^* \mid \theta)} \tag{33}$$

$$= \frac{1}{N} \sum_{i=1}^{N} \delta(\tau, \tau_i^*) \tag{34}$$

Thus, optimal importance sampling simply corresponds to uniformly randomly sampling a data point $i \in [N]$, and then selecting the trajectory $\tau_i^*$. We note that each $\tau_i^*$ simply corresponds to a sequence of actions. Therefore, our sampling procedure corresponds to uniformly randomly selecting sequences from the list $\tau_1^*, \ldots, \tau_N^*$. Additionally, the loss function

gradient is as follows.

$$\frac{d}{d\theta}\log(\Pr(X_1,\dots X_N \mid \theta)) = \frac{d}{d\theta}\log(\prod_{i=1}^{N}\sum_{\tau\in\mathcal{T}}\Pr(X_i \mid \tau)\Pr(\tau \mid \theta)) \tag{35}$$

$$= \frac{d}{d\theta}\sum_{i=1}^{N}\log(\Pr(X_i \mid \tau_i^*)\Pr(\tau_i^* \mid \theta)) \tag{36}$$

$$= \sum_{i=1}^{N}\left[\frac{d}{d\theta}\log(\Pr(X_i \mid \tau_i^*)) + \frac{d}{d\theta}\log(\Pr(\tau_i^* \mid \theta))\right] \tag{37}$$

$$= \sum_{i=1}^{N}\frac{d}{d\theta}\log(\Pr(\tau_i^* \mid \theta)) \tag{38}$$

This is exactly the negative log-likelihood loss. Uniformly randomly selecting sequences (of actions) and then evaluating the negative log likelihood loss on each sequence is simply autoregressive generation. □

**Theorem A.3.** *Since $X_1,\dots,X_N$ only depend on $\theta$ through $S$, the M-step of expectation maximization simplifies to* $\theta^{(t+1)} = argmax_\theta \mathbb{E}_{S\sim\Pr(S\mid X_1,\dots,X_N,\theta^{(t)})}[\log(\Pr(S \mid \theta))]$.

*Proof.* The standard M-step of expectation maximization is as follows.

$$\theta^{(t+1)} = \operatorname*{argmax}_\theta \sum_{i=1}^{N}\mathbb{E}_{S_i\sim\Pr(S\mid X_i,\theta^{(t)})}[\log(\Pr(X_i, S_i \mid \theta))]. \tag{39}$$

Then, we have the following derivation.

$$\theta^{(t+1)} = \operatorname*{argmax}_\theta \sum_{i=1}^{N}\mathbb{E}_{\hat{S}_i\sim\Pr(S\mid X_i,\theta^{(t)})}[\log(\Pr(X_i, \hat{S}_i \mid \theta))]. \tag{40}$$

$$= \operatorname*{argmax}_\theta \sum_{i=1}^{N}\mathbb{E}_{\hat{S}_i\sim\Pr(S\mid X_i,\theta^{(t)})}[\log(\Pr(X_i \mid \hat{S}_i)\Pr(\hat{S}_i \mid \theta))]. \tag{41}$$

$$= \operatorname*{argmax}_\theta \sum_{i=1}^{N}\mathbb{E}_{\hat{S}_i\sim\Pr(S\mid X_i,\theta^{(t)})}[\log(\Pr(\hat{S}_i \mid \theta))] + \mathbb{E}_{S_i\sim\Pr(S_i\mid X_i,\theta^{(t)})}[\log(\Pr(X_i \mid S_i))] \tag{42}$$

$$= \operatorname*{argmax}_\theta \sum_{i=1}^{N}\mathbb{E}_{S_i\sim\Pr(S\mid X_i,\theta^{(t)})}[\log(\Pr(S_i \mid \theta))] \tag{43}$$

$$= \operatorname*{argmax}_\theta N\mathbb{E}_{S\sim\Pr(S\mid X_1,\dots,X_N,\theta^{(t)})}[\log(\Pr(S \mid \theta))] \tag{44}$$

$$= \operatorname*{argmax}_\theta \mathbb{E}_{S\sim\Pr(S\mid X_1,\dots,X_N,\theta^{(t)})}[\log(\Pr(S \mid \theta))]. \tag{45}$$

□

# B. Methodological and Implementation Details

## B.1. Multiple Pools of Observations

**Simple pooled observations:** If multiple pieces of data are generated from each state $S$, one can group together these *sub-observations* into one full observation $X_i$ for each state. For instance, if each $S_i^*$ is a transportation network, each $X_i$ may consist of a list of observed transportation paths (sub-observations) through that network. As the number of sub-observations per observation $X_i$ increases, each observation $X_i$ provides more information about the latent state $S_i^*$, and the task of estimating the latent state $S_i^*$ becomes easier (illustrated in Section 4.1), and the probability distribution $\Pr(X_i \mid S)$ may approach zero for many incorrect states $S \neq S_i^*$. For the PROCESS GRAPH INFERENCE problem in Section 4.1, each state is a directed graph, and observations consist of lists of start and end points of random walks through

**a** $\Pr(X \mid S)$  **b**  **c**  **d**

|       | $S_1$ | $S_2$ | $S_3$ |
|-------|-------|-------|-------|
| $X_1$ | 0.5   | 0     | 0     |
| $X_2$ | 0     | 0.3   | 0.2   |

$S(\tau_1) = S_1$
$S(\tau_2) = S_2$
$S(\tau_3) = S_3$

↑$\Pr(\tau_1 \mid \theta)$ => ↓$r(\tau_1)$

↑$\Pr(\tau_2 \mid \theta)$ => ↓$r(\tau_2)$

$\Pr(\tau_1 \mid \theta) = 0.5$
$\Pr(\tau_2 \mid \theta) = 0.5$
$\Pr(\tau_3 \mid \theta) = 0$

*Figure S1.* **A simple intuitive example applying GReinSS. a** The problem setup is defined by the set of states $S_1, S_2, S_3$, the set of observations $X_1, X_2$, and the probability function $\Pr(X \mid S)$ defined on these states and observations. **b** Policy learning is applied by having the states $S_1, S_2, S_3$ generated by trajectories $\tau_1, \tau_2, \tau_3$. **c** Increasing the probability $\Pr(\tau \mid \theta)$ of either trajectory $\tau_1$ or $\tau_2$, results in a decreased dynamic reward $r(\tau)$ for that trajectory. **d** The dynamic rewards result in an optimal solution where the probabilities $\Pr(\tau_1 \mid \theta)$ and $\Pr(\tau_2 \mid \theta)$ are balanced, maximizing the probability $\Pr(X_{1:N} \mid \theta) = \Pr(X_1 \mid \theta)\Pr(X_2 \mid \theta)$. However, $\Pr(\tau_3 \mid \theta) = 0$ since the trajectory $\tau_3$ is not needed to maximize $\Pr(X_{1:N} \mid \theta)$.

the graph. The number of sub-observations is varied, showing how this impacts the accuracy of each method. In simulations, modifying $n_p$ allows us to test how different methods perform when the amount of information about each ground-truth state $S_i^*$ in each observation $X_i$ varies.

**Multiple environments:** In some cases, one may have several environments $v$, each with their own distribution of latent states $\Pr_v^*(S)$, list of ground-truth latent states $S_{v,1}^*, \ldots S_{v,N^v}^*$, distribution of observations $\Pr_v(X \mid S)$, and list of observations $X_1^v, \ldots X_{N^v}^v$. This may appear to be a generalization, but in fact, it is a special case of the original Problems 2.1 and 2.2. We simply append the environment number $v$ to each observation and state. Specifically, we define $\Pr((X, v) \mid (S, v')) = \Pr_v(X \mid S)$ for $v = v'$, and 0 otherwise. Similarly, define $\Pr^*((S, v)) = \Pr_v^*(S)$, and define $\Pr((S, v) \mid \theta)$ as the probability of the latent state $S$ in environment $v$ given model parameters $\theta$. We then solve Problems 2.1 and 2.2 on the full set of observations $(X_1^1, 1), (X_2^1, 1), \ldots (X_N^M, n_{\text{env}})$ where $n_{\text{env}}$ is the number of environments. This technique is used for modeling RNA splicing in Section 4.2, with each sample treated as an environment.

### B.2. Intuitive Example Demonstrating Adaptive Rewards

Imagine a simple toy problem with three possible states $S_1, S_2, S_3$ and two observations $X_1, X_2$, with $\Pr(X_1 \mid S_1) = 0.5$, $\Pr(X_2 \mid S_2) = 0.3$, $\Pr(X_2 \mid S_3) = 0.2$, and $\Pr(X_i \mid S_j) = 0$ for all other $i \in [2], j \in [3]$ (Figure S1a). For simplicity, imagine each state $S_i$ has its own unique trajectory $\tau_i$ (Figure S1b). The fixed reward function without GReinSS's rescaling is $r'(\tau) = \sum_{i=1}^N \Pr(X_i \mid \tau)$. Thus, $r'(\tau_1) = 0.5$, $r'(\tau_2) = 0.3$, $r'(\tau_3) = 0.2$. Since $\tau_1$ achieves the highest reward, standard policy gradients would solve for the optimal solution where $\Pr(\tau_1 \mid \theta) = 1$, $\Pr(\tau_2 \mid \theta) = \Pr(\tau_3 \mid \theta) = 0$. Thus, $\Pr(X_2 \mid \theta) = 0$, and $\Pr(X_1, X_2 \mid \theta) = 0$.

For GReinSS we have the following rewards.

$$r(\tau_1) = \frac{\Pr(X_1 \mid \tau_1)}{\Pr(X_1 \mid \tau_1)\Pr(\tau_1 \mid \theta)} = \frac{1}{\Pr(\tau_1 \mid \theta)} \tag{46}$$

$$r(\tau_2) = \frac{\Pr(X_2 \mid \tau_2)}{\Pr(X_2 \mid \tau_2)\Pr(\tau_2 \mid \theta) + \Pr(X_2 \mid \tau_3)\Pr(\tau_3 \mid \theta)} \tag{47}$$

$$r(\tau_3) = \frac{\Pr(X_2 \mid \tau_3)}{\Pr(X_2 \mid \tau_2)\Pr(\tau_2 \mid \theta) + \Pr(X_2 \mid \tau_3)\Pr(\tau_3 \mid \theta)} \tag{48}$$

We have $r(\tau_2) > r(\tau_3)$ for any $\theta$ which results in $\Pr(\tau_3 \mid \theta) = 0$. Thus, we have the following simplification.

$$r(\tau_2) = \frac{\Pr(X_2 \mid \tau_2)}{\Pr(X_2 \mid \tau_2)\Pr(\tau_2 \mid \theta)} = \frac{1}{\Pr(\tau_2 \mid \theta)} \tag{49}$$

When $\Pr(\tau_1 \mid \theta) > \Pr(\tau_2 \mid \theta)$ then $r(\tau_2) > r(\tau_1)$ and when $\Pr(\tau_1 \mid \theta) < \Pr(\tau_2 \mid \theta)$ then $r(\tau_2) < r(\tau_1)$ (Figure S1c). The equilibrium solution is $r(\tau_1) = r(\tau_2)$ when $\Pr(\tau_1 \mid \theta) = \Pr(\tau_2 \mid \theta)$. Thus, GReinSS solves for $\Pr(\tau_1 \mid \theta) = \Pr(\tau_2 \mid \theta) = 0.5$, and $\Pr(\tau_3 \mid \theta) = 0$ (Figure S1d). Thus, $\Pr(X_1 \mid \theta) = 0.5^2 = 0.25$, and $\Pr(X_2 \mid \theta) = 0.5 \cdot 0.3 = 0.15$. Consequently, $\Pr(X_1, X_2 \mid \theta) = 0.25 \cdot 0.15 = 0.0375$.

## B.3. Methodological comparison of GReinSS with expectation maximization

GReinSS and generalized expectation maximization (GEM) both have the goal of maximizing the probability $\Pr(X_1, \ldots, X_N \mid \theta)$. GEM consists of alternating between an E-step of estimating the latent states and an M-step of optimizing $\theta$ to maximize an expectation value given the estimated latent states. Specifically, for Problem 2.1, GEM cannot be solved exactly, so the E-step is approximated by predicting latent states $\hat{S}_1, \ldots, \hat{S}_N$ with $\hat{S}_i = \mathrm{argmax}_S \Pr(S \mid \theta) \Pr(X_i \mid S)$. Then, since $X_i$ only depends on $\theta$ through $S$, the M-step simplifies to optimizing $\theta$ to maximize the $\sum_{i=1}^{N} \log(\Pr(\hat{S}_i \mid \theta))$ as shown in Theorem A.3.

With this approximation, the first step of GReinSS and GEM both consist of sampling $S$ from $\Pr(S \mid \theta)$. For GReinSS, the next step is using $\Pr(X_i \mid S)$ to calculate rewards for the sampled states and modifying $\theta$ using policy gradients based on these rewards. For GEM, the next step is using $\Pr(X_i \mid S)$ to estimate $\hat{S}_1, \ldots \hat{S}_N$ using the sampled states and then modifying $\theta$ to maximize the probability $\sum_{i=1}^{N} \log(\Pr(\hat{S}_i \mid \theta))$. Both GReinSS and GEM repeatedly alternate between these steps until convergence of $\sum_{i=1}^{N} \log(\Pr(\hat{S}_i \mid \theta))$. Since the M-step of GEM consists of maximum likelihood generative modeling, it can utilize any machine learning method designed for this task, including autoregressive models, variational autoencoders, and discrete diffusion.

## B.4. Prior Applications of GReinSS

Two previous papers have utilized special cases of the GReinSS technique without formulating the full generalized GReinSS procedure (Ivanovic & El-Kebir, 2023; 2025). One of these papers introduces CLoMu, a method for modeling and predicting cancer clonal evolution (Ivanovic & El-Kebir, 2023). CloMu specifically defines the latent states as unobserved phylogenetic (evolutionary) trees representing the clonal evolution of SNV mutations in the tumor of a specific patient. These phylogenetic trees are generated by starting with an unmutated cell, and iteratively adding SNV mutations to populations of cells in order to form a phylogeny tree of populations of cells with different mutations. Thus, the actions generating the trajectory correspond to adding mutations to populations of cells (clones) in order to form new populations of cells (clones) with new mutations. The combinatorial structure of phylogeny trees makes their generation via policy learning very natural. Phylogeny trees are never directly observed. Instead, DNA sequencing enables noisy measurements of mutations on existing populations of cells, from which sets of possible phylogeny trees can be derived. The observations used consist of possible sets of phylogeny trees for each patient, and are generated from the DNA sequencing data.

The second prior application of GReinSS is CNRein (Ivanovic & El-Kebir, 2025), a method for inferring CNV mutations on individual cells from single-cell DNA sequencing of tumors. A latent state consists of the set of CNV mutations present in an individual cell in the form of a copy number profile that indicates the total number of copies of each region of the genome, as well as the haplotype that the copies correspond to. Mathematically, each latent state is represented by a sequence of pairs of integers in which each integer in the pair represents one of the two haplotypes. These latent states are generated by iteratively adding CNV mutations to a cell, starting with a normal cell. Thus, actions correspond to selecting a CNV mutation to add, represented in terms of the genomic region covered by the CNV, the haplotype of the CNV, and the copy number change of the CNV. The observation data consist of a processed form of the DNA sequencing data of each cell. One component of this sequencing data is the read depth, which is defined as the number of DNA sequencing reads that are determined to originate from each genomic region. Another component is the B-allele frequency, which is defined as the proportion of DNA sequencing reads that are determined to originate from each haplotype for some given genomic region. Mathematically, each of these is represented by a list of real values with one real value for each genomic position. Although these measurements are biologically complex, the relationship between these observations and the latent states is modeled with a simple Gaussian distribution. For instance, the read depth of a genomic position (a component of $X_i$) is modeled by a Gaussian distribution with a mean equal to the integer total copy number of that genomic position (a component of $S$). This formulation is very similar to our latent set reconstruction problem, with the modification of allowing for integer-valued vectors rather than binary vector representations.

## B.5. Latent State Inference

Given some model parameters $\theta$, and the ability to sample states from the distribution $\Pr(S \mid \theta)$, we describe the procedure for solving Problem 2.2 and predicting states $\hat{S}_1, \ldots, \hat{S}_N$ with the goal of matching the ground-truth latent states $S_1^*, \ldots, S_N^*$. Let $\mathbb{I}_{S=S'}$ be defined as 1 if $S = S'$ and 0 otherwise. Note that $\Pr(X_i \mid S) \Pr(S \mid \theta) = \mathbb{E}_{S \sim \Pr(S \mid \theta)}[\mathbb{I}_{S=S'} \Pr(X_i \mid S')]$ can be estimated via sampling from $\Pr(X_i \mid S)$ for all methods regardless of whether $\Pr(X_i \mid S)$ has a closed-form expression. Thus, for all methods we sample $s$ from $\Pr(S \mid \theta)$ and predict the state $\hat{S}_i = \mathrm{argmax}_{S'} \mathbb{E}_{S \sim \Pr(S \mid \theta)}[\mathbb{I}_{S=S'} \Pr(X_i \mid S')]$.

For methods with off-policy sampling, we apply the importance sampling correction to this expectation value. During the training of GEM-based methods, we sample a batch size of 1000 latent states to calculate the E-step estimates of $\hat{S}_1, \ldots, \hat{S}_N$ prior to the M-step update of model parameters $\theta$. For all methods, during the final prediction of latent states, we sample 100,000 latent states from $\Pr(S \mid \theta)$ to calculate $\hat{S}_1, \ldots, \hat{S}_N$.

### B.6. Importance Sampling

After modifying the sampling procedure, one simply has to use importance sampling to account for this in policy gradient. Let $\pi'$ be the modified sampling procedure that utilizes the observations $X_1, \ldots, X_N$ to generate trajectories $\tau$ with high probabilities $\Pr(X_i \mid \tau)$. Then, applying importance sampling gives the below equation

$$\frac{d}{d\theta} \log(\Pr(X_1, \ldots X_N \mid \theta)) = \mathbb{E}_{\tau \sim \pi'}[r(\tau) \frac{\Pr(\tau \mid \theta)}{\Pr(\tau \mid \pi')} \frac{d}{d\theta} \log(\Pr(\tau \mid \theta))]. \tag{50}$$

Thus, off-policy sampling of $\tau$ gives the correct policy gradient if one includes the importance sampling correction $\Pr(\tau \mid \theta)/\Pr(\tau \mid \pi')$.

### B.7. Model Architectures

All models utilize relatively similar neural network architectures, so that the differences in results between models result from their different training procedure rather than their model architecture. All policy-learning-based methods, including GReinSS, naive policy gradients, and trajectory balance GFlowNets, use an identical model architecture. This model consists of a two-layer fully connected neural network with 50 hidden neurons and the PyTorch leakyRelu non-linearity. The autoregressive model uses a similar architecture consisting of a two-layer neural network with autoregressive masking and consequently uses a number of hidden neurons equal to the number of tokens in the input. The variational autoencoder uses two two-layer neural networks with 50 hidden neurons for both the encoder and decoder. The size of the latent representation inside the auto-encoder is also 50. The discrete diffusion model also uses a two-layer neural network with 50 hidden neurons. For the discrete diffusion model, 20 timesteps are used. Timestep values are concatenated with the input after first applying a cosine-based transformation. Specifically, the timestep $t$ is embedded into a 20 dimensional vector where the $i$th element is defined as $\cos(t \cdot i \cdot \pi/20)$. The local search algorithm does not use a trained model. Latent states are instead represented as a binary vector, and for each $X_i$ each iteration changes one element in the binary vector in order to maximize $\Pr(X_i \mid s)$.

### B.8. Additional Baseline Details

The autoregressive model is trained using the standard negative log-likelihood loss. The variational autoencoder is trained using the standard evidence lower bound (ELBO) loss. The discrete diffusion model is trained using the standard denoising diffusion loss. Trajectory balance GFlowNets use the standard trajectory balance loss. Naive policy gradients uses standard policy gradients with no augmentation of the reward function. GReinSS and GEM-based approaches are trained until $\Pr(X_1, \ldots, X_N \mid \theta)$ converges. Naive policy gradients are trained until the achieved reward converges, and GFlowNets are trained until the loss function converges. The local search algorithm iterates through the observations $X_1, \ldots, X_N$ and for each observation $X_i$ it searches for the state $S$ that maximizes $\Pr(X_i \mid S)$. For each $X_i$, local search starts with a binary vector representation of the latent state $S$ and performs a sequence of modifications to this vector until reaching a locally optimal state $S$. Specifically, each modification selects the single element modification to $S$ that maximizes $\Pr(X_i \mid S)$. This continues until convergence of $\Pr(X_i \mid S)$ such that no single element modification to $S$ can improve $\Pr(X_i \mid S)$. For the set reconstruction simulations, local search is equivalent to including an element in the state $S$ if and only if the value for that element in the observation is greater than 0.5. Consequently, local search is guaranteed to find the globally optimal $S$ for maximizing $\Pr(X_i \mid S)$ for the set reconstruction simulations.

## C. Additional Experimental Details

### C.1. Process Graph Inference Simulation Details

In each graph-based simulation, latent states correspond to graphs that are generated by applying weight thresholding on some base graph. Each simulation instance has a randomly generated directed base graph with each directed edge being included with probability $1/2$. Weights for each directed edge in the base graph are uniformly randomly generated from $U(\frac{1}{4}, 1)$ (with the lower limit set above 0 to ensure all edges in the base graph actually occur in some latent graph).

Each of the 1000 graphs in a simulation is generated by uniformly randomly generating a threshold from $U(0,1)$ and then performing weight thresholding (Yan et al., 2018) on the base graph with that threshold.

Observations consist of the start and end points of random walks in the graph. Specifically, the walk begins by uniformly randomly selecting the start node. Then at each step we randomly choose between each edge directed out of that node as well as the "stop" action of ending on the current node. The stopping action and each outgoing edge are chosen with equal probability. Note that this implies if the current node has no outgoing edges then it will always be stopped on if reached. Each $X_i$ consists of the list of start and end points of all of the random walks generated. The number of random walks generated is set to 10, 100, and 1000 in the three simulations.

## C.2. $F_1$ Score Calculation

The $F_1$ score is defined as follows

$$F_1 = 2\frac{\text{precision} \cdot \text{recall}}{\text{precision} + \text{recall}} = \frac{2 \cdot \text{true positives}}{(2 \cdot \text{true positives}) + \text{false positives} + \text{false negatives}}. \tag{51}$$

Let $\hat{S}_i$ be a predicted state and $S_i^*$ be the true latent state, where latent states represent either sets or graphs as in our simulations (Section 4.1). Then, true positives consist of elements or edges in both $\hat{S}_i$ and $S_i^*$. False positives consist of elements or edges in $\hat{S}_i$ but not $S_i^*$. False negatives consist of elements or edges in $S_i^*$ but not $\hat{S}_i$. From these three quantities, an $F_1$ score is calculated for any predicted latent state $\hat{S}_i$ given the true latent state $S_i^*$. The distribution of $F_1$ scores for the $N$ predictions $\hat{S}_{1:N}$ is either summarized by median values as in Figures 2 and 3, or visualized fully as in Figures S3, S5 and S6.

## C.3. Set Reconstruction Simulation Details

The latent states correspond to sets of elements represented by binary vectors of size $v_{\text{size}}$. The default size of the vectors is 100, but additional modified simulations use vectors of size 10 and 1000. The sets are generated by taking the union of subsets of elements in a dictionary of reusable modules (Mairal et al., 2009). The number of subsets included in each state and the number of elements in each subset are both set to scale with $\sqrt{v_{\text{size}}}$ so that the proportion of elements included in each state remains roughly constant. Specifically, the dictionary contains $\sqrt{v_{\text{size}}}$ (rounded down) subsets/modules. Each subset is generated by randomly including each element with probability $2/\sqrt{v_{\text{size}}}$. Each state is generated by randomly including each subset in the dictionary with probability 0.1. As an example with $v_{\text{size}} = 10$, if the dictionary contains the two subsets $\{2, 3, 5, 7\}$ and $\{1, 2, 3, 4\}$, and the state includes both of these subsets, then the state would consist of $\{1, 2, 3, 4, 5, 7\}$ which is equivalent to the binary vector $[1, 1, 1, 1, 1, 0, 1, 0, 0, 0]$. The observation would then consist of adding Gaussian noise with mean 0 and standard deviation $\sigma$ to the vector $[1, 1, 1, 1, 1, 0, 1, 0, 0, 0]$.

## C.4. Off-policy Sampling Procedure for Simulated Set Reconstruction

For generating each trajectory $\tau$, the off-policy sampling procedure first, with 50% probability, either samples a trajectory on-policy or selects a random observation $X_i$ from the set of observations $X_1, \ldots, X_N$. If an observation $X_i$ is selected, we use the following procedure. Let $X_{i,1}, \ldots, X_{i,M} \in \mathbb{R}$ represent the observed values for each of the $M$ elements in the universe. For any state $S$, let $S_j \in \{0, 1\}$ represent whether or not the $j$th element is included in the state. Intuitively, if $X_{i,j}$ is large ($> 0.5$) we wish to increase the probability that $S_j = 1$, and if $X_{i,j}$ is small ($< 0.5$) we wish to decrease the probability that $S_j = 1$. The value $\sigma$ is the standard deviation of the Gaussian noise in the simulation. Define $P_G(i, j)$ as the log probability ratio of $S_j$ taking the value 1 rather than the value 0 given $X_{i,j}$, which is equal to $\frac{1}{2\sigma^2}\left(X_{i,j}^2 - (1 - X_{i,j})^2\right)$ since $X_{i,j}$ comes from a Gaussian with standard deviation $\sigma$ centered at either 0 or 1. Given the state $S$ and model parameters $\theta$ let $P_O(S, j, \theta)$ be the logits proportional to the log probability of the next action being adding the element $j$ to the set $S$. The new off-policy logits (proportional to the log probability of the next action adding state $j$ to set $S$) are then $P_O(S, j, \theta) + P_G(i, j)$. The softmax operation then converts these logits to probabilities. Intuitively, the probability of adding the element $j$ to the set $S$ increases proportionally to $P_G(i, j)$, which is the log probability ratio that $S_j = 1$ rather than $S_j = 0$ given $X_{i,j}$.

## C.5. Generative Model for RNA Splicing

Junctions are defined as the connection between distinct exons in an isoform and can be represented by tuples containing the ending genomic position of one exon and the starting genomic position of the following exon. For instance, if an isoform contains the exon ranging from the genomic position 1000 to 1100 followed by the exon ranging from genomic position 2000 to 2100, then the junction connecting these exons can be represented by $(1100, 2000)$. We choose to represent isoforms in terms of sequences of junction tuples. Note that the GTEx read count data with stable genomic positions is only publicly available in terms of the number of reads covering each junction. The exact nucleotides in an isoform can be precisely defined in terms of a sequence of junctions, with the minor exception of the exact length of the first and last exons. We model isoforms via a generative process, starting with an empty isoform and iteratively selecting junctions to include in the isoform. Since human data on GTEx has known possible exons, we allow our model to select its next junction $j$ if the start position of this junction $j$ and the end position of the previously selected junction $j'$ define a known exon. For species without known exons, a reasonable approximation of this is to allow the model to select its next junction $j$ if the start position of $j$ is within a certain number of nucleotides (corresponding to a reasonable exon length) of the end of the previous junction. The first junction is allowed to be any junction starting in the end position of any starting exon in the GENCODE v39 annotation. The isoform is allowed to end on a junction if the junction's ending position is the start position of any ending exon in the GENCODE v39 annotation. For species without known exons, one could approximate by allowing starting and ending positions within a certain distance of the lowest and highest genomic positions observed in the reads mapped to the relevant gene. The probability of each of the possible junctions at each step is determined by a neural network from an input of a one-hot encoding of the junctions already selected in the isoform. The neural network is specifically a two-layer neural network with 50 hidden neurons, and a leakyReLU non-linearity.

To avoid ambiguity, we use the term *sequencing sample* to refer to a specific sequencing aliquot on GTEx, which is the unit with individual read count data. This is distinct from tissue samples from which one can derive multiple sequencing aliquots, and distinct from our policy learning sampling procedure. After generating the isoform, we select the probability of this isoform for each sequencing sample in the dataset, and the probability of observing reads from each junction in the isoform. Selecting the sequencing sample is performed using neural network with the same architecture as is used for constructing the isoform. Selecting the probability of reads from each junction is done by applying a learned sequencing bias vector with one log probability for each junction. The logit for each junction in the isoform is set to this sequencing bias value, and the logit for each junction not in the isoform is set to approximately negative infinity (technically, the bias minus 500, corresponding to a probability below $e^{-500}$ to avoid numerical issues). The log probability of each junction is then calculated via a log softmax.

Although isoforms are sampled according to the policy, off-policy sampling is used for selecting the sequencing sample and junction that the read comes from. Specifically, we simultaneously generate all possible sequencing samples and junctions in order to reduce reward variance during training. We therefore use the importance sampling correction term in our loss function.

## C.6. RNA Splicing Error Calculation

We calculate the isoform prediction error using the Jaccard distance between isoforms as well as optimal transport to account for isoform proportions. This section will first describe the general calculation together with an illustrative example, before moving on to technical details for the GTEx dataset. The Jaccard index for measuring similarity between two sets is defined as one minus the size of the intersection of the two sets divided by the size of the union of the two sets. The Jaccard distance is then one minus the Jaccard index. For genomic regions, the Jaccard distance is then one minus the ratio of the number of nucleotides in the intersection of the two regions to the number of nucleotides in the union of the two regions. A hypothetical example is shown in Figure S7a. In this example, isoform 1 consists of an exon from 100bp to 200bp, an exon from 300bp to 400bp, and an exon from 500bp to 550bp, while isoform 2 consists of an exon from 150bp to 250bp and an exon from 300bp to 400bp. Then, the intersection is 150bp to 200bp as well as 300bp to 400bp, totaling 150 nucleotides. The union is 100bp to 250bp, 300bp to 400bp, and 500bp to 550bp, totaling 300bp. Thus, the Jaccard distance in this example is $1 - (150/300) = 0.5$. For each tissue sample, there can be multiple isoforms with different proportions detected by long-read sequencing, and multiple isoforms with different proportions predicted from short-read sequencing by either GReinSS or RSEM. Figure S7b shows a hypothetical example where long-read sequencing detects 80% proportion to isoform 1 and 20% proportion to isoform 2, whereas the prediction assigns 60% proportion to isoform 1 and 40% proportion to isoform 2. To account for the proportions of each isoform on each sample, we calculate a Jaccard distance matrix between all pairs of predicted isoforms and long-read detected isoforms, and then apply optimal transport to these proportions. This is

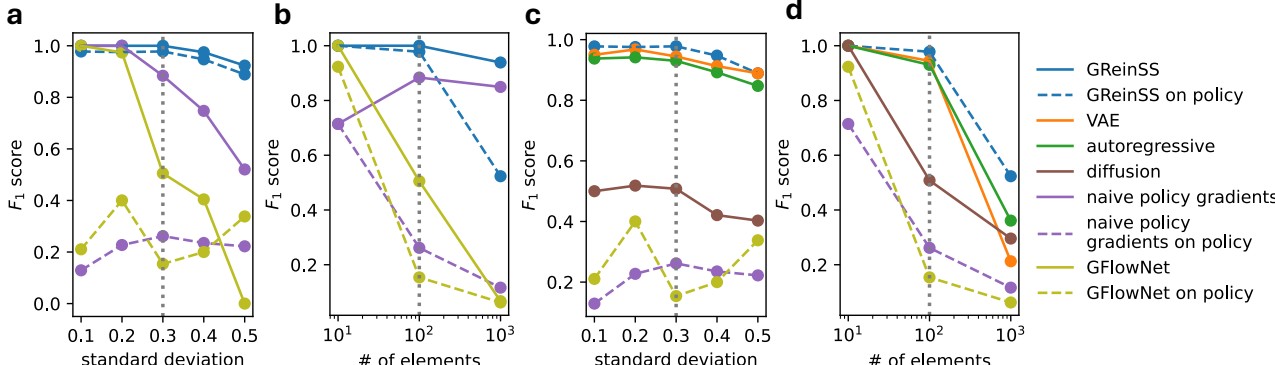

*Figure S2.* **Ablation of policy learning methods on the set reconstruction simulation to remove off-policy sampling. a-b** Policy learning methods consistently achieve a higher performance when off-policy sampling is used. Panel **a** has varying standard deviations $\sigma$ and panel **b** has varying sizes of the universe of elements. **c-d** Off-policy GReinSS consistently outperforms all off-policy and non-RL machine learning baselines. Panel **c** has varying standard deviations $\sigma$ and panel **d** has varying sizes of the universe of elements.

illustrated for our hypothetical example in Figure S7c. Specifically, the distance matrix contains $0.5$ as the distance between isoform 1 and isoform 2, and contains the distance 0 between each isoform and itself. Optimal transport then assigns $20\%$ proportion to the prediction and long-read agreeing on isoform 1, $60\%$ proportion to the prediction and long-read agreeing on isoform 2, and $20\%$ proportion to the prediction having isoform 2 but long-read having isoform 1. The $20\%$ proportion of the prediction having isoform 2 but long-read having isoform 1 results in an error of $20\%$ of the Jaccard distance between isoform 1 and isoform 2, namely $0.1 = 0.5 \cdot 20\%$. In general, optimal transport yields a correspondence between predicted isoforms and long-read isoforms that minimizes the error. As a special case, if there is only one long-read detected isoform and only one predicted isoform, optimal transport simplifies to the Jaccard distance between these two isoforms. After calculating errors on each tissue sample with optimal transport, we calculate a weighted average of errors across tissue samples. Specifically, each sample is weighted by the long-read sequencing read count (of reads covering whole isoforms). This weighting is essential since some tissue samples have very few or even zero long-read sequencing reads (covering whole isoforms).

**Additional details:** Long read sequencing directly detects entire isoforms through reads that cover entire isoforms. Consequently, we filter for reads covering entire isoforms when calculating long-read sequencing read counts. As another technical detail, the GTEx database can have multiple sequencing aliquots for each tissue sample that have their own RNA sequencing data. To compare isoform estimations between short-read sequencing data and long-read sequencing data, we first pool the predicted isoform quantities across sequencing aliquots for each tissue sample. Then, for the same set of 61 tissue samples, we have isoforms predicted from short-read sequencing data from GReinSS and RSEM, as well as isoforms quantified from long-read sequencing using FLAIR. As discussed in Section C.5, we define isoforms in terms of their list of junctions. Therefore, we define the Jaccard distance with the region of an isoform defined as including all exons between the first and last junctions (i.e., only excluding the starting and ending exons themselves). We then calculate the Jaccard distance between all pairs of predicted isoforms and the isoforms estimated from long-read sequencing to form a distance matrix used in the optimal transport error calculation.

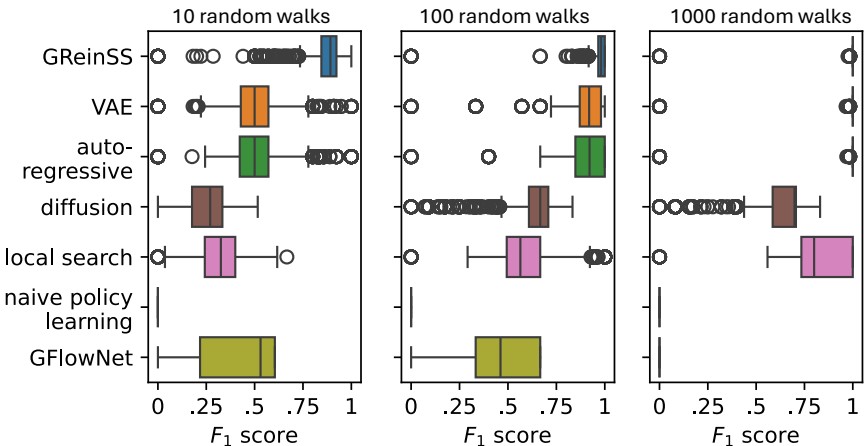

*Figure S3.* **The distribution of $F_1$ scores of each method on graph inference simulations with** $10$**,** $100$ **and** $1000$ **random walks per observation.** The difficulty increases for smaller numbers of random walks, resulting in a larger advantage for GReinSS.

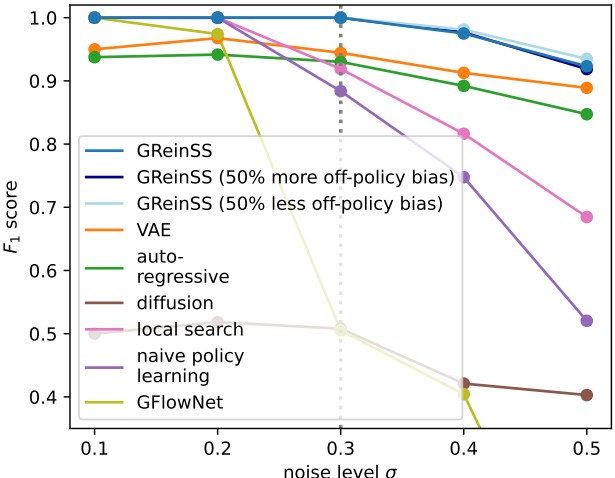

*Figure S4.* **GReinSS is not sensitive to changes in the off-policy sampling proposal.** We tested GReinSS's sensitivity to modifying the off-policy sampling proposal by artificially increasing and decreasing the strength of the sampling bias on the action logits by $50\%$. There was very little impact on the $F_1$ score (with a maximum decrease of $0.0044$ for any $\sigma$ value), with GReinSS remaining the top performing method.

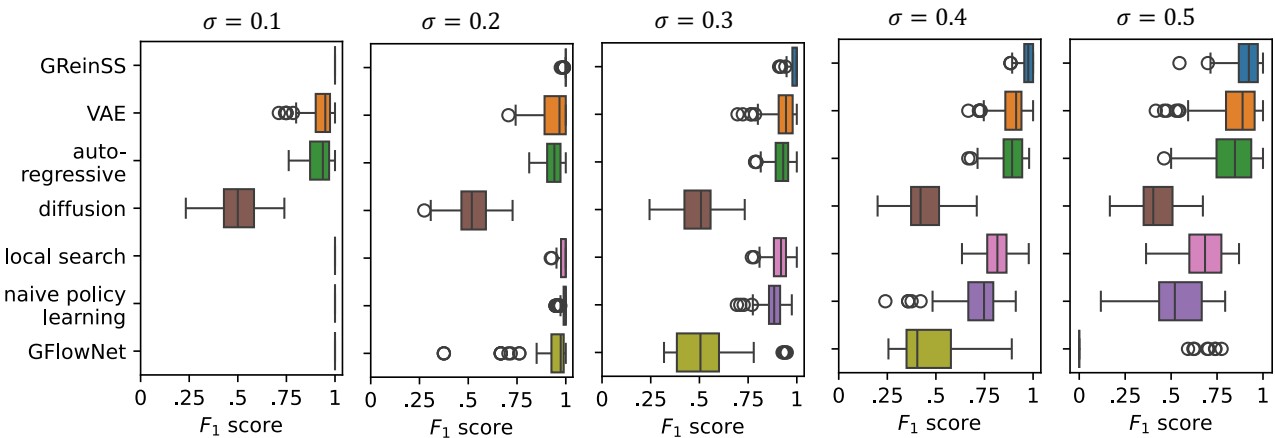

*Figure S5.* **The distribution of $F_1$ scores of each method on set inference simulations with standard deviations $\sigma$ set to $0.1$, $0.2$, $0.3$, $0.4$, and $0.5$.** The difficulty increases as the variance increases, but GReinSS maintains the top performance.

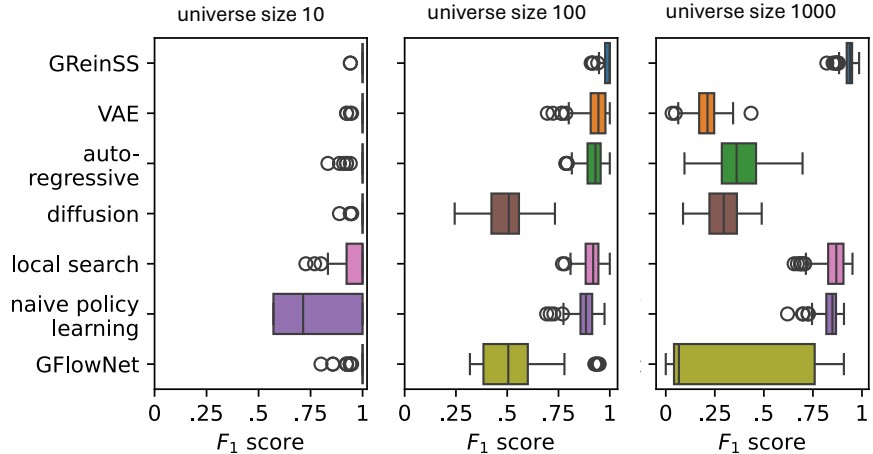

*Figure S6.* **The distribution of $F_1$ scores of each method on set inference simulations with the size of the universe of elements set to $10$, $100$ and $1000$.** The difficulty increases as the universe of elements increases in size, but GReinSS maintains the top performance.

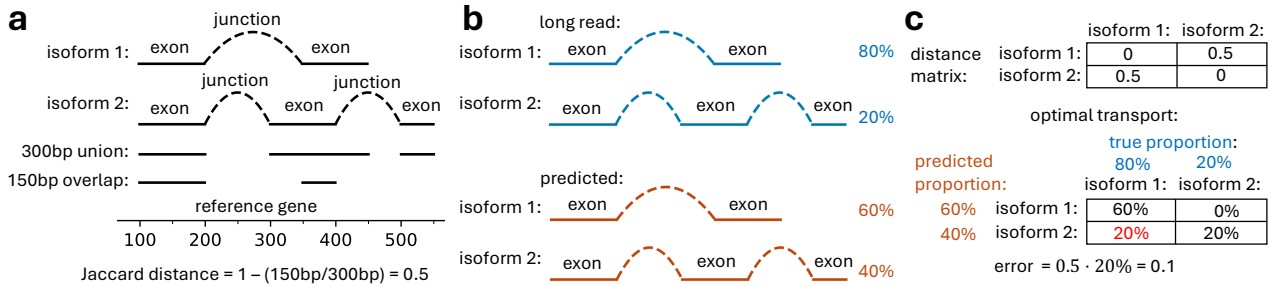

*Figure S7.* **A hypothetical example of calculating the isoform prediction error. a** Isoform 1 consists of an exon from 100bp to 200bp and an exon from 350bp to 450bp. Isoform 2 consists of an exon from 100bp to 200bp, an exon from 300bp to 400bp, and an exon from 500bp to 550bp. The intersection is 150bp, and the union is 300bp, resulting in a Jaccard distance of $1 - (150/300) = 0.5$. **b** In this example, the long-read sequencing has isoform 1 with proportion $80\%$ and isoform 2 with proportion $20\%$, while the prediction has isoform 1 with proportion $60\%$ and isoform 2 with proportion $40\%$. **c** The error is calculated using optimal transport. Specifically, the prediction has $20\%$ more of isoform 2 that should be predicted as isoform 1, resulting in an error of $20\%$ of the Jaccard distance of $0.5$ between isoform 1 and 2, namely $0.1$.

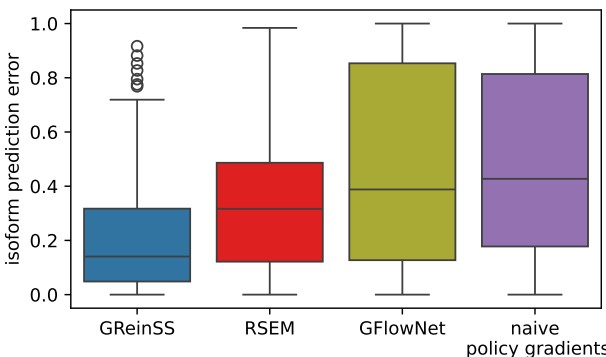

*Figure S8.* **GReinSS outperforms GFlowNets and naive policy gradients in the isoform reconstruction task.** GFlowNets and naive policy gradients were run on 100 random genes for the RNA isoform reconstruction task. For these genes, the median isoform prediction error is 0.141, 0.317, 0.388, and 0.427 for GReinSS, RSEM, GFlowNets, and naive policy gradients, respectively. The poor performance of GFlowNets and naive policy gradients clearly indicates that a neural network reparameterization of isoform generation is insufficient to achieve high accuracy, and the GReinSS approach is necessary to achieve this improvement.

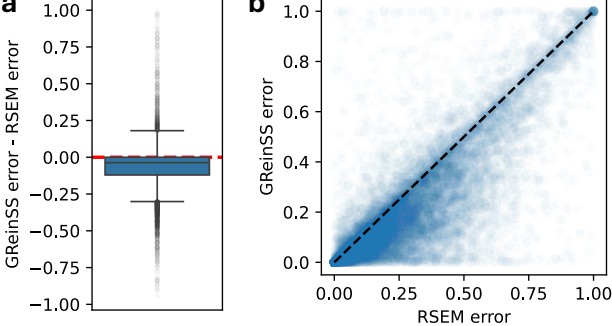

*Figure S9.* **Additional comparisons of GReinSS and RSEM errors in isoform proportion prediction. a** The full boxplot of the GReinSS error minus the RSEM error, including all outliers. **b** A scatterplot showing the GReinSS error and RSEM error for all genes.

