# OpenReview forum: "Generative Modeling of Discrete Latent Structures via Dynamic Policy Gradients"
_ICML.cc/2026/Conference — ICML 2026 regular_

### Official Review · Reviewer_TqwY · 2026-03-08

**Soundness:** 4
**Presentation:** 3
**Significance:** 3
**Originality:** 2
**Overall Recommendation:** 5
**Confidence:** 2

**Summary:**

The authors propose a new method that generalizes two previous published applications (Ivanovic & El-Kebir, 2023 and 2025), called GReinSS, that aims to learn models of the type Prod_i Sum_S p(X_i|S)p(S|theta), where S is a latent variable that is discrete and combinatorial in nature (such as evolutionary trees, graphs, sets and so on). The parameters of p(X_i|S) are assumed known. Using concepts derived from the framework of reinforcement learning, they use a policy approach, where a policy sequentially generates latent structures, and build upon an unbiased estimator of the gradient of the log-likelihood objective log(p(X_1:N | theta)) that is estimated using Monte Carlo. Evaluation includes simulated data (graph inference, subset inference) and an application to RNA-seq.

**Compliance With Llm Reviewing Policy:**

Affirmed.

**Final Justification:**

The rebuttal clarified some remaining questions wrt novelty and scope, and I updated my rating accordingly. This seems to be an interesting method to deal with discrete latent states, with potential wide application.

**Key Questions For Authors:**

1. Relationship to prior work (Appendix B.4)
The main article and the appendix notes that two earlier papers by the authors already used closely related techniques in two biological applications. Could the authors clarify why this follow-up article is at the level of an ICML publication? What is specifically novel in the paper apart from generalization and clarification, especially since only one additional non-synthetic application is given (and again in biology)?

2. Sensitivity to off-policy sampling strategies
The paper discusses the importance of off-policy sampling and heuristic proposals for approximating the optimal sampling distribution. How sensitive are the results to the specific heuristics used to construct these proposal distributions? How easy it is too come up with these heuristics?

3. Generality beyond the presented tasks.
The method is presented as a general framework for latent structured inference. Could the authors comment on how easily the approach transfers to other domains beyond the examples studied here, particularly settings where the latent structures or likelihood models differ substantially? What are the conditions for applying this method?

4. What if the parameters of p(X|S) are NOT known? This is an important limitation.

**Limitations:**

yes

**Strengths And Weaknesses:**

Soundness:
The paper is (as far as I can see) technically sound. The derivation connecting the dynamic reward to the gradient of the marginal likelihood follows standard score-function and policy-gradient arguments. The assumptions are reasonable for problems where the observation model is known and latent structures are generated sequentially. The experiments include two synthetic tasks (graphs, sets) and one real biological application (RNA-Seq) with several baselines, including EM-style and policy-learning methods. While the results support the claims, the evaluation is somewhat limited and relies heavily on synthetic settings.

Presentation:
The paper is generally clear and well structured. The sections on problem formulation, method, and experiments are logically organized, and the figures are helpful. The paper discusses related approaches such as EM, variational inference, and policy learning, though the relationship to closely related prior work could be highlighted earlier (see below).

Significance:
The paper studies an important problem: learning distributions over combinatorial latent structures from indirect observations. Such problems appear in several scientific domains, especially computational biology. The RNA splicing example demonstrates potential practical relevance. However, the overall impact may be somewhat specialized. The method requires a known observation model and sequential generation of latent structures, which may limit broader applicability.

Originality: The article builds on several wel-known ideas, though brought together in a creative way to solve a timely problem. However, the authors write on page 2, lower right: "In addition to related general methods, two previous papers have utilized the underlying technique of GReinSS without generalizing to describe the full GReinSS procedure (Ivanovic & El-Kebir, 2023; 2025)." Thus, the novelty of the work is a bit doubtful to me. The core idea (using policy-gradient updates with likelihood-based rewards to infer discrete and combinatorial latents) appears in earlier domain-specific work by the same authors. The main contribution of this paper (I gather) is to abstract and formalize this approach into a general framework (GReinSS), provide a theoretical justification linking the reward to the likelihood gradient, and evaluate the method on two artifical data sets and one additional biological problem.

---

> ### Author Rebuttal · Authors · 2026-03-30
>
> **Q1 relation to prior works:** We agree that the two prior works CloMu and CNRein use policy gradient methods with reward rescaling, which can be described as domain-specific special cases of GReinSS. However, compared to the current manuscript, these works **did not**: (i) formulate a **general probabilistic learning and inference problems** on arbitrary discrete structures (Section 2), (ii) provide **theoretical grounded solution** for these problems by establishing a connection between policy-gradient training and maximum likelihood estimation in discrete combinatorial spaces (Section 3.1) (iii) formulate a **theory for optimal off-policy sampling** (Section 3.2) rather than using purely ad-hoc binary restrictions on sampling (as done in prior works), (iv) **unify several existing ML approaches** as special cases and approximations of GReinSS (Section 3.3), or (v) provide **ablations and comparisons** with existing ML methods (Section 4).
>
> Specifically, both CloMu and GReinSS use an evolutionary model of the temporal progression of cancer with actions corresponding to the addition of mutations, and apply domain-specific reward scaling to account for noise. However, their policy learning trajectories are **tightly coupled to biologically temporal mutation trajectories, rather than serving as a general method** for generating unobserved combinatorial states. Finally, their off-policy sampling consisted entirely of using ad-hoc restrictions on sampling by disallowing certain actions, with no theoretical grounding for how this improves training.
>
> In summary, the five essential contributions listed above allow GReinSS to be widely applied across domains in future work as well as allow for future improvements on the framework itself. We plan to revise our manuscript to better emphasize these contributions and the limitations of the two prior works.
>
> **Q2 clarification on off-policy sampling proposals:**  Only the set inference simulation results use off-policy sampling, whereas the RNA results and graph simulation results do not. As shown in our response to Reviewer “iaEU” (Q1), **GReinSS is not sensitive to the specific heuristics used for off-policy sampling**, as long as the sampling bias is generally directionally correct. Off-policy sampling heuristics are easy to construct since they only need to downweight implausible actions given the observations. For example, the off-policy sampling introduced in the set inference simulation uses a simple Gaussian probability, which is trivial to implement. In general, if the observations strongly suggest an action should not occur,  the proposal should bias against it. The off-policy proposal does not need to closely match the theoretical optimum proposal (as shown in response to Q2 of Reviewer "iaEU").
>
> **Q3 flexibility of GReinSS to other applications:** We thank the reviewer for bringing up this important component for the value of the GReinSS framework. **GReinSS transfers easily across problems with very different latent structures and likelihood models**, as long as basic requirements (e.g., an approximately known $\Pr(X|S)$) are met. The key advantage is the very high flexibility of policy learning, enabling highly distinct and complex generative processes.
>
> The tradeoff for this flexibility is the standard challenges of policy gradient optimization. Thus, GReinSS is most useful under conditions where simpler approaches are insufficient. For example, if the states are generally known, direct maximum likelihood generative modeling may be preferable (as we will mention in the discussion section of the revised manuscript).
>
> **Q4 unknown observation generating probabilities :** We agree that GReinSS does not apply when $\Pr(X|S)$ is completely unknown. However, when $\Pr(X|S)$ is partially unknown, a practical approach is to factor the unknown aspects of $\Pr(X|S)$ into $\Pr(S|\theta)$.
>
> For example, in the RNA setting, the process of generating sequencing reads $X$ from isoforms $S$ is unknown (and non-uniform across the genome), giving an apparent difficulty for GReinSS. **We address this by including read position information in the latent state S**, so $\Pr(S|\theta)$ captures the read-generating process, and $\Pr(X|S)$ becomes a simple indicator (0 or 1), depending on whether the state $S$ includes the read position of $X$ or not.
>
> More generally, unknown aspects of $\Pr(X|S)$ can be incorporated into $\Pr(S|\theta)$ by **augmenting the latent state $S$ with information $X$**, at the cost of increased latent state complexity. As noted in the discussion, future work could instead explicitly parameterize $\Pr(X|S)$. We thank the reviewer for pointing out this important topic, which highlights not only challenges but also techniques GReinSS implementations can use to address these challenges. We will highlight this latent state augmentation technique in the revised manuscript.

---

> > ### Author Rebuttal · Reviewer_TqwY · 2026-04-03
> >
> > I am satisfied with the response and will update my rating accordingly.

---

> > > ### Author Response · Authors · 2026-04-06
> > >
> > > We are glad we addressed your concerns, appreciate your insightful questions and suggestions, and greatly appreciate your update to the rating!

---

### Official Review · Reviewer_CRnh · 2026-03-10

**Soundness:** 3
**Presentation:** 3
**Significance:** 3
**Originality:** 3
**Overall Recommendation:** 5
**Confidence:** 3

**Summary:**

The paper studies latent-state modeling from indirect observations where latent states S are discrete/combinatorial (graphs, sets, isoforms) and observations X are generated via a known (or computable) likelihood. The goal is to learn a generative model maximizing the marginal likelihood. The core contribution is GReinSS, which casts sampling of S as terminal states of trajectories, and derives a policy-gradient estimator using a dynamically rescaled reward. They claim this yields an unbiased estimator of the log-likelihood gradient. Experiments cover (1) simulated directed-graph inference from random-walk endpoints, (2) simulated subset inference from noisy per-element measurements, and (3) RNA isoform reconstruction from short-read data with evaluation against long-read isoforms; GReinSS is reported to outperform a range of baselines including naive policy gradients, GFlowNets, local search, and GEM-style training with VAE/AR/diffusion components.

**Compliance With Llm Reviewing Policy:**

Affirmed.

**Final Justification:**

The rebuttal addressed my concerns, and the authors promised to add more info on minibatching in the final manuscript.

**Key Questions For Authors:**

* What is the computational overhead of GReinSS relative to naive policy gradients and GEM? Specifically, how does the cost of estimating $Pr(Xᵢ | θ)$ for all i scale?
* Do you approximate the reward sum over i using minibatches? If so, is the resulting update optimizing a stochastic approximation to the full-data likelihood or something else?
* Can you add an empirical convergence analysis (e.g., log-likelihood curves over training)?

**Limitations:**

yes

**Strengths And Weaknesses:**

Strength:

* The method is well-motivated and well-written. The reward rescaling is an intuitive “balancing” mechanism that prevents collapse to a single high-reward trajectory and instead encourages covering all observations.
* The discussion of when the framework reduces to standard MLE modeling, local search, standard policy gradients, or when GFlowNets may solve the same objective is useful for positioning.
* The paper includes both controlled simulations and a realistic bioinformatics application (isoform reconstruction) where the supervision is genuinely indirect and evaluation uses an orthogonal measurement modality (long-read).

Weakness:

*  The paper does not provide any theoretical or empirical analysis of gradient variance. In practice the paper estimates the reward by Monte Carlo sampling,  which can exhibit high variance when $Pr(Xᵢ | θ)$ is small.
*  The simulations use relatively small state spaces (graphs on 10 nodes, universes up to 1000 elements, N=1000 observations). It is unclear how GReinSS scales with (a) the size of the latent state space |S|, (b) the number of observations N, and (c) the number of samples M used to estimate $Pr(Xᵢ | θ)$.
* The only baseline for the RNA isoform task is RSEM. The paper does not compare against other isoform quantification tools (e.g., Salmon, kallisto, StringTie) or against the GEM/GFlowNet/naive policy gradient baselines used in the simulations. This makes it difficult to assess whether the gains come from the GReinSS framework specifically or from the neural network parameterization of the isoform generative model.
* Corollary 3.2 states that GReinSS performs gradient ascent on the log-likelihood, but there is no discussion of convergence properties. Standard policy gradient convergence results assume fixed rewards. With dynamic rewards estimation (Monte carlo) that depend on θ, it is not obvious that the iterative procedure converges, or at what rate.

---

> ### Author Rebuttal · Authors · 2026-03-30
>
> **Q1 computation: GReinSS’s computational overhead relative to naive policy gradients is minimal**, requiring only estimating observation probabilities and adjusting rewards. For either method, the main computational cost is generating trajectories and policy gradient updates. Computing $\Pr(X_i | \theta)$ primarily consists of computing $\Pr(S | \theta)$, which is already required for standard policy gradients. The computational cost of GReinSS relative to GEM depends on how the states $S$ are generated, since this dominates the runtime for both methods. For instance, autoregressive models require sequential generation, while VAEs generate states in parallel. Overall, VAEs and local search are the fastest due to their simpler state generation.
>
> **Q2 minibatching:**  We did not use minibatches for computing the reward sum over $i$, since reward computation is cheap relative to trajectory generation and policy gradient updates. For RNA data with up to millions of observations, we aggregate observations into count matrices to avoid individually computing reward contributions for each observation.
>
> For future settings where this is infeasible, minibatching may be required, as we will note in the discussion section. As suggested by the reviewer, the resulting update optimizes a stochastic approximation to the full data likelihood. The policy gradient decomposes over observations, with the contribution from each observation only depending on the set of states generated and not the other observations. Thus, minibatching yields an unbiased stochastic gradient of the full objective.
>
> **Q3 convergence:** Thank you for the helpful suggestion for demonstrating GReinSS’s smooth practical convergence. We generated log-likelihood curves as suggested, showing **stable training across different numbers of random walks** ($k \in 10, 100,1000$) for the graph inference simulations.
>
> k=10:  https://imgur.com/a/J9g1RnC
>
> k=100:  https://imgur.com/a/DL2XLsl
>
> k=1000:  https://imgur.com/a/gy0z2aH
>
>
>
> **Addressing weaknesses:**
>
> **Variance:** We agree that Monte Carlo sampling exhibits higher variance than non-sampling methods, however, all ML baselines also rely on random sampling. Additionally, GReinSS reduces variance via reward scaling; specifically, each observation contributes exactly 1.0 to the total reward in a given batch, so trajectory rewards roughly reflect the number of well-fit observations rather than an arbitrarily scaled raw probability value. The reward is bounded between 0 and the number $N$ of observations, but in practice is on the scale of a small subset of the number of observations. Empirically, convergence plots show that the variance is low enough for smooth training (shown in Q3).
>
> **Scaling:** The runtime is dominated by trajectory generation and gradient updates, which have costs proportional to the number of actions required to generate each trajectory, the action computation time, and the number $M$ of trajectories in each batch. The size $|\mathcal{S}|$ of the state space is roughly exponential in the trajectory length, so computational costs are roughly logarithmic in $|\mathcal{S}|$. The number $N$ of observations impacts the reward computation, which is comparatively computationally cheap. Even for very large $N$ (e.g., RNA-seq), aggregating observations into count matrices enables efficient reward computation. However, exploring GReinSS usage on much larger datasets/problems is an important direction for future work (which we will add to the discussion section).
>
> **RNA:** To determine whether gains on the RNA application come from the GReinSS framework specifically or from the neural network parameterization of the isoform generative model, we implemented the naive policy gradients and GFlowNets ablations on the RNA application and ran them on 100 randomly selected genes. These ablations perform worse than either GReinSS or RSEM, with the median isoform prediction error being 0.141, 0.317, 0.388, and 0.427 for GReinSS, RSEM, naive policy gradients, and GFlowNets, respectively, with boxplots shown in https://imgur.com/a/O6zSguJ . This confirms that **the gains come specifically from the GReinSS framework** and not just the neural network parameterization of isoform generation.
>
>
> **Convergence:** We completely agree that standard policy gradients convergence relies on fixed rewards, and thus does not apply to our setting. GReinSS performs stochastic gradient ascent on a fixed log-likelihood objective using an unbiased gradient estimator. Standard stochastic optimization results, therefore, imply convergence to a stationary point given sufficiently small step sizes. While we do not provide theoretical guarantees on the rate of convergence (which also depends on neural net topology), the variance of the gradient estimator is controlled in practice since the rewards are bounded between $0$ and $N$, contributing to the smooth empirical convergence observed in our likelihood curves (see Q3).

---

> > ### Author Rebuttal · Reviewer_CRnh · 2026-04-03
> >
> > I thank the authors for the detailed and well-structured rebuttal. The responses address my key concerns convincingly. I want to flag that the discussion around minibatching is, in my view, especially important for the paper's broader applicability. The current approach of aggregating observations into count matrices works well for the RNA setting, but as the authors acknowledge, this will not be feasible in all settings. The connection between minibatching and optimizing a stochastic approximation to the full data likelihood is a valuable clarification, and I strongly encourage the authors to include this discussion prominently in the revision, as it directly speaks to how practitioners in other domains might adopt the framework.
> > Overall, the rebuttal has strengthened my confidence in the paper's contributions. I will maintain my score of 5 (accept).

---

> > > ### Author Response · Authors · 2026-04-06
> > >
> > > We completely agree, and thank the reviewer for emphasizing this important addition to our manuscript for helping practitioners apply GReinSS to their domains. We will add a prominent discussion of minibatching to the revised manuscript.

---

### Official Review · Reviewer_iaEU · 2026-03-10

**Soundness:** 3
**Presentation:** 3
**Significance:** 3
**Originality:** 3
**Overall Recommendation:** 4
**Confidence:** 3

**Summary:**

This paper introduces GReinSS, a framework for generative modeling of discrete mechanistic latent states from indirect observations. The setting is one where the latent states are combinatorial and unobserved, but the observation model is known or partly known. The main idea is to train a policy over latent states using dynamically rescaled rewards so that the policy-gradient update is an unbiased estimator of the observed-data log-likelihood gradient. The paper gives the main likelihood-gradient result, an off-policy result, and experiments on simulated latent graph inference, simulated latent set reconstruction, and RNA isoform reconstruction from short-read sequencing data.

**Compliance With Llm Reviewing Policy:**

Affirmed.

**Final Justification:**

The rebuttal addressed my remaining concerns and I would like to change my recommendation to an "accept".

**Key Questions For Authors:**

1. The practical method relies heavily on heuristic off-policy proposals. How sensitive are the results to those choices across the synthetic tasks and the RNA application?
2. The paper derives the variance-minimizing off-policy proposal. Do the authors have evidence on how performance changes when the implemented proposal is a poor approximation to that optimum?
3. On the RNA isoform task, could the authors clarify why RSEM is the main comparison point and whether additional practical baselines are available?

**Limitations:**

Yes

**Strengths And Weaknesses:**

Strengths: A clear strength of the paper is that it does not rely on a proxy reinforcement-learning target or on artificial latent variables that are easier to learn but no longer correspond to the states of interest. The main theorem is therefore the key contribution: it shows that the dynamically rescaled reward leads to an unbiased policy-gradient estimator of the observed-data likelihood gradient. The empirical section is also well matched to the claim. The two synthetic tasks are useful because the true latent structures are known, so the paper can test whether the method actually recovers the intended objects rather than only fitting the observations. Those experiments are convincing and show strong gains over several baseline families, including policy-learning baselines and EM-style generative baselines. The RNA isoform experiment is also a meaningful real-data case study because evaluation is done using orthogonal long-read sequencing rather than only an internal training objective.

Weaknesses: The practical pipeline relies heavily on heuristic off-policy proposals. Although the paper derives the variance-minimizing proposal, the implemented method uses domain-specific approximations, so part of the empirical performance depends on sampling choices whose robustness across domains is not yet clear. Second, the originality claim should be framed carefully. The paper is clear that earlier domain-specific works already used special cases of the technique; the contribution here is to unify the framework, sharpen the theory, and evaluate it across settings. That is still meaningful, but it is different from introducing the core idea from scratch. Third, the real-data evidence is more limited than the synthetic evidence. The RNA isoform experiment is relevant, but it is still a single real application, so the strongest support for the method comes from the synthetic tasks where the true latent structures are known, but this is not a major problem by any means.

---

> ### Author Rebuttal · Authors · 2026-03-30
>
> **Q1 clarification on off-policy proposal:** The heuristic **off-policy proposal was only used for the set inference simulations and was not used for either the graph inference simulations or the RNA application**. GReinSS is not generally reliant on heuristic off-policy proposals; however, our set inference simulation was included to demonstrate how off-policy proposals can improve the performance for certain specific problems.
>
> This being said, we agree that moving away from heuristics in off-policy sampling proposals would be valuable. As mentioned in the discussion, we intend to improve upon off-policy GReinSS in future work by including a learned off-policy sampling procedure that is automatically trained towards matching the optimal off-policy sampling procedure.
>
> Additionally, we performed new experiments (using the set inference simulations) and found that the results are **not very sensitive to the precise off-policy sampling proposal chosen**, as long as the off-policy sampling proposal generally biases sampling towards trajectories that match observations. Specifically, we artificially increased and decreased the strength of the sampling bias on the action logits by 50%, and found **almost no impact** on the $F_1$ score (decreasing by less than 0.0044), with GReinSS clearly remaining the top performing method (plotted in https://imgur.com/a/86HrHtZ ).
>
> **Q2 approximations of optimal off-policy proposals:** This insightful question is very valuable for understanding practical applications of GReinSS, given the challenges of implementing exactly optimal off-policy sampling. As mentioned in response to Q1, using a weak approximation of the optimal off-policy sampling proposal, for instance, by artificially increasing or decreasing the bias, has a very small impact on the results. More generally, it is worth noting that our implemented off-policy sampling proposal for the set inference problem is not optimal, since it is highly non-trivial to estimate the optimal proposal. In our discussion, we mentioned the valuable future direction of directly training an off-policy sampling model to match the optimal off-policy sampling proposal. We are currently investigating this future work approach and gathering initial promising results.
>
>
> **Q3 RNA application baselines:** The GTEx database does not make the raw data required to run alternative isoform quantification methods publicly available. However, since RSEM is very standard in the field (Conesa et al. Genome biology 2016), GTEx has run RSEM on its private raw data and made the outputs of RSEM publicly available. **This allows us to compare with a highly used standard method (RSEM) on GTEx, without having access to private raw data**. GReinSS was run on publicly available junction read counts, rather than the private raw data. We have also now **run the naive policy gradients and GFlowNets ablations** of GReinSS on 100 random genes, and found that they **perform worse than both GReinSS and RSEM**. Specifically, on these 100 random genes, the median isoform prediction error is 0.141, 0.317, 0.388, and 0.427 for GReinSS, RSEM, naive policy gradients, and GFlowNets, respectively, as shown in this boxplot https://imgur.com/a/O6zSguJ.
>
>
>
> **Addressing remaining weakness:**
> We agree that it is important to emphasize the prior applications of CloMu and CNRein. While these biological application-specific methods were essential in the path of ideas that led to the GReinSS framework, both of **these methods were entirely focused on their particular biological problem rather than introducing a widely applicable learning framework.** In fact, these papers specifically define their policy as following the temporal structure of cancer evolution, rather than viewing policy learning as a general technique for generating discrete data. Both papers also focus entirely on the biological validation of their outputs and evolutionary model, rather than the underlying properties of their reinforcement learning technique and how it could apply to domains outside of cancer evolution. Finally, **the two papers do not include a theoretical framework, nor a comparison against ML baselines** (as further elaborated in response to Reviewer TqwY). Nevertheless, CloMu and CNRein have played an essential role in demonstrating that policy gradients with rescaled rewards can effectively solve real-world computational biology problems. Additionally, as special cases of GReinSS, they provide strong evidence for the effectiveness of GReinSS on real data problems beyond our RNA application. We will make sure to emphasize this in the updated manuscript.

---

> > ### Author Rebuttal · Reviewer_iaEU · 2026-04-02
> >
> > The rebuttal addressed my primary concerns and clarified the role of the heuristic off-policy proposal. The authors provided useful evidence that the method is not very sensitive to that choice in the setting where it is used and clarified the rationale for the RNA baseline while adding additional ablations. The response on prior related work also makes the intended contribution much clearer. The issues I raised were adequately addressed.

---

> > > ### Author Response · Authors · 2026-04-06
> > >
> > > Thank you for your positive assessment and feedback. We are glad we could fully address your concerns, clarifying the usage of off-policy learning, analyzing the sensitivity to the off-policy proposal, adding additional baselines, and clarifying our major contributions. We will update the manuscript to reflect your valuable feedback.

---

### Official Review · Reviewer_Rehw · 2026-03-24

**Soundness:** 3
**Presentation:** 2
**Significance:** 3
**Originality:** 3
**Overall Recommendation:** 5
**Confidence:** 3

**Summary:**

This paper introduces GReinSS, a policy gradient framework for learning distributions over discrete latent states from indirect observations. The core insight is a dynamically rescaled reward function (Theorem 3.1) that yields unbiased log-likelihood gradient estimates. The paper formally unifies GFlowNets, EM/GEM, VAEs, autoregressive models, and standard policy gradients as special cases, with each reduction rigorously proven. Experiments on graph inference, subset inference, and RNA isoform reconstruction demonstrate consistent improvements over baselines.

**Compliance With Llm Reviewing Policy:**

Affirmed.

**Final Justification:**

The rebuttal confirmed my initial assessment of the paper.

**Key Questions For Authors:**

None

**Limitations:**

Yes.

**Strengths And Weaknesses:**

Strengths

- The dynamic reward formulation (Theorem 3.1) providing unbiased log-likelihood gradients provides useful insight, and its proof is clean.
- Thorough experiments covering three qualitatively different latent structure types: directed graphs, subsets inference, and RNA isoforms.
- The ablation showing baseline methods fail at k=10 (Figure 2b) directly validates the theoretical contribution of the dynamic reward rescaling.

Weaknesses

- No wall-clock runtime comparisons are provided. If GReinSS is significantly slower than GEM-based baselines or RSEM, the practical significance would be reduced, particularly for the RNA application where RSEM is used at scale.

---

> ### Author Rebuttal · Authors · 2026-03-30
>
> Thank you for the accurate summary and positive review!
>
> Regarding the one mentioned weakness, **the computational requirements of GReinSS are very low**: We re-ran GReinSS on the RNA application for a random sample of 100 genes (for all patients) on a MacBook Pro (Apple M2 Max) without the use of a GPU and found a median runtime of 11.95 seconds per gene (mean runtime 15.42 seconds) with a boxplot shown at https://imgur.com/a/v39TXGi. The GTEx database does not provide the raw data required to run RSEM and instead only provides the final output, making it impossible to provide an exact runtime comparison with RSEM. However, the very low computational requirements of GReinSS make it very quick and effective to apply at scale.

---

> > ### Author Rebuttal · Reviewer_Rehw · 2026-04-06
> >
> > I would like to thank the authors for their response. My concerns have been fully addressed, and I find the paper both interesting and its contributions solid. Based on the authors’ response and the feedback from other reviewers, I am now more confident in my assessment.

---

> > > ### Author Response · Authors · 2026-04-06
> > >
> > > We are glad that we could fully address your concerns, and we thank the reviewer for considering the paper’s contributions as both interesting and solid. We are also happy to have increased the strength and confidence of the reviewers' positive acceptance assessment.

---

### Decision · Program_Chairs · 2026-04-30

**Decision:**

Accept (regular)

**Comment:**

The submission introduces a general methodology to infer discrete latent structures based on policy gradient, which unifies previousl approaches. All reviewers agree on the significance of the contribution, despite the moderate novelty of the approach. The experiments are judged convincing and interesting, as they cover a variety of discrete structures and explore relevant real data.